# Insight into mode-of-action and structural determinants of the compstatin family of clinical complement inhibitors

Christina Lamers [1,2], Xiaoguang Xue[3], Martin Smieško [1], Henri van Son [3], Bea Wagner[1], Nadja Berger [4], Georgia Sfyroera[4], Piet Gros [3,5], John D. Lambris[4,5] ✉ & Daniel Ricklin [1,5] ✉

With the addition of the compstatin-based complement C3 inhibitor pegcetacoplan, another class of complement targeted therapeutics have recently been approved. Moreover, compstatin derivatives with enhanced pharmacodynamic and pharmacokinetic profiles are in clinical development (e.g., Cp40/AMY-101). Despite this progress, the target binding and inhibitory modes of the compstatin family remain incompletely described. Here, we present the crystal structure of Cp40 complexed with its target C3b at 2.0-Å resolution. Structure-activity-relationship studies rationalize the picomolar affinity and long target residence achieved by lead optimization, and reveal a role for structural water in inhibitor binding. We provide explanations for the narrow species specificity of this drug class and demonstrate distinct target selection modes between clinical compstatin derivatives. Functional studies provide further insight into physiological complement activation and corroborate the mechanism of its compstatin-mediated inhibition. Our study may thereby guide the application of existing and development of next-generation compstatin analogs.

The human complement system contributes to various pathologies, from autoimmune, age-related, and inflammatory disorders to transplant- and biomaterial-induced complications, making it a prime target for therapeutic intervention[1,2]. Despite growing interest, the development of complement-targeted drugs has been slow, with two related anti-C5 antibodies (eculizumab, ravulizumab) approved for the treatment of paroxysmal nocturnal hemoglobinuria (PNH) and other indications long remaining the only clinical options[2]. It was only in 2021, with the FDA and EMA approval of pegcetacoplan (Empaveli®/Aspaveli®, Apellis), when a second class of complement inhibitors with distinct mechanism became available[3,4]. Compared to existing therapies, pegcetacoplan acts upstream in the complement cascade by impairing the activation of the central component C3 to provide a broader control of complement effectors. An extension of therapeutic intervention points within the cascade has been highly anticipated in view of the diverse involvement of complement in pathology.

Complement primarily acts as a rapid host defence system that eliminates microbial intruders and apoptotic cells[1]. After initiation by various means, including immune complexes (classical pathway) or microbial signatures (lectin pathway), the cascade converges at the activation of the plasma protein C3 by convertases (Fig. 1a). C3 cleavage releases the anaphylatoxin C3a and produces an opsonic fragment (C3b), which covalently attaches to the target cell surface. Concerted binding of the proteases factor B (FB) and factor D (FD) to C3b generates the main C3 convertase (i.e., C3bBb) to activate more C3

[1]Department of Pharmaceutical Sciences, University of Basel, Klingelbergstrasse 50, 4056 Basel, Switzerland. [2]Institute of Drug Discovery, Faculty of Medicine, Leipzig University, Brüderstrasse 34, 04103 Leipzig, Germany. [3]Department of Chemistry, Faculty of Science, Utrecht University, Padualaan 8, 3584 Utrecht, The Netherlands. [4]Department of Pathology & Laboratory Medicine, Perelman School of Medicine, University of Pennsylvania, 401 Stellar Chance, 422 Curie Blvd, Philadelphia 19104 PA, USA. [5]These authors jointly supervised this work: Piet Gros, John D. Lambris, Daniel Ricklin.
✉e-mail: lambris@pennmedicine.upenn.edu; d.ricklin@unibas.ch

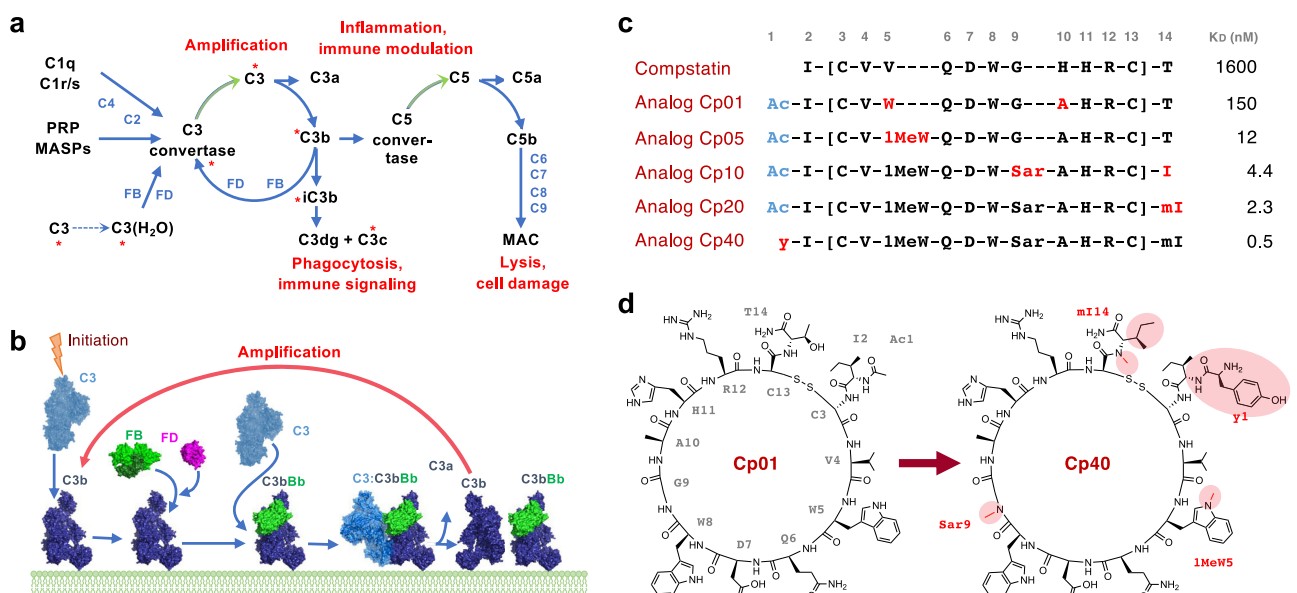

**Fig. 1 | Overview of complement mechanisms and compstatin-based inhibitor development. a** Simplified scheme of the complement cascade with major effector functions. Molecular targets of compstatin are marked with an asterisk. **b** Structural model of the amplification loop based on the crystal structures of C3 (2A73)[22], C3b (2I07)[18], FB (2OK5)[50], FD (2XW9)[5], C3b₂Bb₂SCIN₂ (2WIN)[6], and C3a (4HW5)[51]. **c** Amino acid sequences and target affinities of compstatin and major analogs relevant for this study. Residue numbers are indicated at the top, and changes from the previous analog are highlighted in red. Binding affinities (K_D) to C3/C3b determined by surface plasmon resonance (SPR) are shown. **d** Comparison between the chemical structures of analogs Cp01 and Cp40, wherein modifications are highlighted in red. Abbreviations: Ac, Acetyl; FB, factor B; FD, factor D; MAC, membrane attack complex; MASPs, mannose binding lectin-associated serine proteases; PRP, pattern recognition protein. Reprinted from[7] Trends in Pharmacological Sciences, Vol 8, C. Lamers, D.C. Mastellos, D. Ricklin, J.D. Lambris, Compstatins: the dawn of clinical C3-targeted complement inhibition, 629–640, Copyright 2022, with permission from Elsevier.

(Fig. 1a)[5,6]. In absence of regulators, this process feeds an amplification loop (alternative pathway) that rapidly opsonizes surfaces with C3b (Fig. 1b). While C3b and its degradation fragments are directly involved in phagocytic and adaptive immune signaling, C3b also provides a platform for the formation of C5 convertases. Cleavage of C5 produces the inflammatory mediator C5a and generates membrane attack complexes (MAC) that lyse or damage susceptible cells (Fig. 1a). Whereas these potent effector functions provide an important layer of antimicrobial defence, any excessive or misguided complement activation may drive clinical complications by inducing tissue damage, inflammation and adverse immune reactions[1]. Depending on the disorder, a pathway- or effector-specific inhibition may prove sufficient, while other conditions require an approach that suppresses complement activity more broadly[2].

The compstatin family of C3 inhibitors, to which pegcetacoplan belongs, is particularly suited for broad complement inhibition as it largely impairs convertase-mediated C3 activation by all pathways and prevents most effector generation[7]. Compstatin was originally derived from phage display as a disulfide-bridged, 13-amino-acid peptide with micromolar binding affinity for C3[8,9] that was optimized for improved affinity, efficacy, and pharmacokinetic properties (Fig. 1c)[7]. Substituting residues in the cyclic core with proteinogenic (analog Cp01; Fig. 1c, d) and non-proteinogenic amino acids (analog Cp05; Fig. 1c) profoundly enhanced target affinity[7,10,11]. Compstatin Cp05 builds the base for pegcetacoplan, in which two Cp05 units are bridged by a 40-kDa PEG moiety to reduce renal elimination[12]. Finally, backbone N-methylation (Cp10, Cp20)[13,14] and the addition of D-Tyr to the N-terminus yielded analog Cp40 (Fig. 1c, d), which featured picomolar affinity and an improved pharmacokinetic profile in absence of PEGylation. Cp40 was successfully evaluated in preclinical models for PNH, transplantation, and periodontal disease, among others, and is currently in clinical development (AMY-101, Amyndas) for various indications, including periodontitis and acute respiratory distress syndrome in COVID-19 patients[4,7,15,16].

Despite major achievements in developing the compstatin drug class, important aspects such as the molecular determinants of target binding and species specificity, and even the precise mode-of-action, have not been fully explored. Currently, structural insights into the compstatin family and previous structure-activity-relationship (SAR) studies are primarily based on the structure of a first-generation analog (i.e., Cp01) bound to a functionally less relevant C3 fragment (i.e., C3c)[17]. A detailed insight into the binding and inhibitory modes of current analogs is considered critical for future development efforts.

In this study, we solved the crystal structure of the clinical candidate Cp40 with its target C3b and performed SAR studies to identify the key determinants for target binding and selectivity. We also investigated the mode-of-action by elucidating the effect of Cp40 on the formation and activity of C3 convertases. Finally, we compared the binding modes of mono- and bivalent compstatin analogs to elucidate the impact on activity profiles. These insights are immediately relevant for the development of novel complement inhibitors to extend therapeutic options for a wide variety of pathologies, but also enhance our understanding of the molecular mechanisms of complement activation.

## Results

### Compstatin analogs share a target binding site
We crystallized Cp40 with plasma-purified C3b based on our experiences with Cp01-C3c and various (co)-crystal structures of C3b[5,6,17–21]. The crystals diffracted to 2.0 Å, improving on the 2.4 Å of the previous structure[17] despite the larger size of C3b versus C3c (i.e., 175 vs. 135 kDa). Compared to recent C3b-containing structures, the complement C1r/C1s, Uegf, Bmp1 (CUB) domain was less defined, as visible in a low electron density and high B-factors for this domain. All other regions were well defined (Supplementary Fig. 1).

As expected, the protein was organized in the macroglobulin (MG) 1–6^β and linker (LNK) domains of the β-chain, and the α'NT, MG6^α −8, anchor, and C-terminal complement (CTC) domains of the α-chain that are all shared between C3c and C3b[18,22]. Instead of the truncated

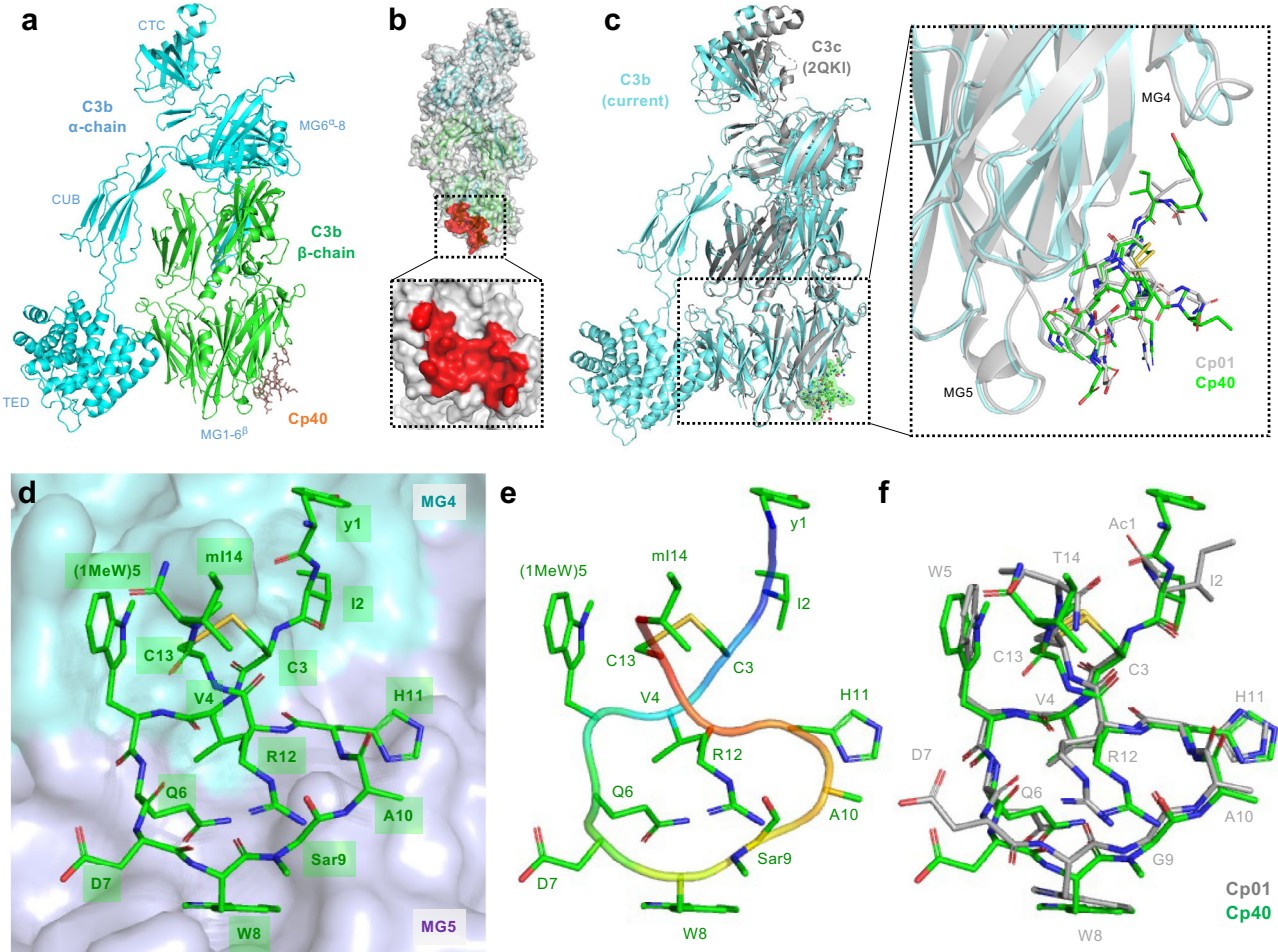

**Fig. 2 | Crystal structure of Cp40-C3b complex and comparison with Cp01-C3c structure. a** Cp40-C3b structure solved at 2.0-Å resolution showing a single molecule of Cp40 (stick representation) binding to the β-chain of C3b (green, cartoon representation; the α-chain is shown in cyan). **b** Cp40 binding site at the MG core of C3b highlighted in red. **c** Superimposition of the current Cp40-C3b structure (green/cyan) with the previous Cp01-C3c structure (grey). The dotted box shows an enlarged view of the compstatin binding region at the interface of the MG4 and MG5 domains of C3b/C3c. The two compstatin analogs are shown as sticks. **d**–**f** Structure of Cp40 in its target-bound conformation. **d** Cp40 is bound as cyclic 14-amino-acid peptide (green stick representation) engaging in intermolecular contacts with the MG4 (cyan) and MG5 (purple) domains of C3b. **e** Backbone trace (main chain as cartoon, side chain as sticks) revealing a twisted, O-shaped bound conformation of the disulfide-bridged peptide **f** Superimposition of Cp01 (grey; from C3c-Cp01 complex) and Cp40 (green; from C3b-Cp40 complex) showing an overall structural similarity with deviations primarily found in side chains not engaging in tight intermolecular interactions. CTC C-terminal complement domain, CUB complement C1r/C1s, Uegf Bmp1 domain, MG macroglobulin domain, TED thioester-containing domain.

CUB domain in C3c, the current structure had an intact CUB and associated thioester domain (TED), as characteristic for C3b (Fig. 2a)[18]. The peptide Cp40 was located at a shallow site formed by domains MG4 and MG5 of C3b (Fig. 2a, b).

In the superimposition of the Cp01-C3c and Cp40-C3b structures, the shared domain regions of C3c and C3b are arranged comparably with only the CTC domain deviating notably (Fig. 2c). The latter was reported to be comparatively flexible due to its separation from the core by an anchor domain[20]. Importantly, the compstatin binding sites are identical between the two proteins (Fig. 2c, closeup).

### Cp40 gains target binding via intra- & intermolecular bonds and shielding of structural water

The structure of C3b-bound Cp40 is well defined (Fig. 2d, e) with clear electron density and acceptable B-factor values (Supplementary Fig. 2). It shows a similar conformation to that of Cp01-C3c (Fig. 2f), with a cyclic structure being enabled by a disulfide bridge between Cys3 and Cys13 (Fig. 2d). In line with previous observations[14,17], target-bound Cp40 does not adopt the open conformation described for compstatin in solution[23] but is square-shaped with a perpendicular disulfide bridge

(Fig. 2e and Supplementary Fig. 3). Superimposition of the target-bound structures of Cp01 and Cp40 revealed subtle differences between the cyclic cores, which mainly affected the side chain orientations of the charged amino acids Asp7, Arg12 and, to a minor degree, His11. Residues that were previously identified as essential for target binding (Trp5, Trp8, Val4, Gln6) deviate little (Fig. 2f). The N-methyl of sarcosine (Sar9) is oriented toward bulk solvent with no significant contact to C3b. We observed profound differences for the residues flanking the ring, which was expected due to extensions and amino acid substitutions at the termini. The crystal structure confirms that D-Tyr1 interacts with a hydrophobic pocket on the MG4 domain, thereby extending the compstatin binding site (Figs. 2d and 3a). Surprisingly, four of the 14 amino acids in Cp40 (Asp7, Ala10, Arg12, mIle14) do not directly engage with the target in the crystal structure (Supplementary Table 1).

For quantitative assessment of the contacts of target-bound Cp40, we employed molecular dynamics (MD) simulations to generate trajectories for calculating the binding energies based on molecular mechanics combined with generalized Born and surface area continuum solvation (MM/GBSA method)[24]. Furthermore, we focused on per-residue contributions from hydrogen bonds (H-bonds) and

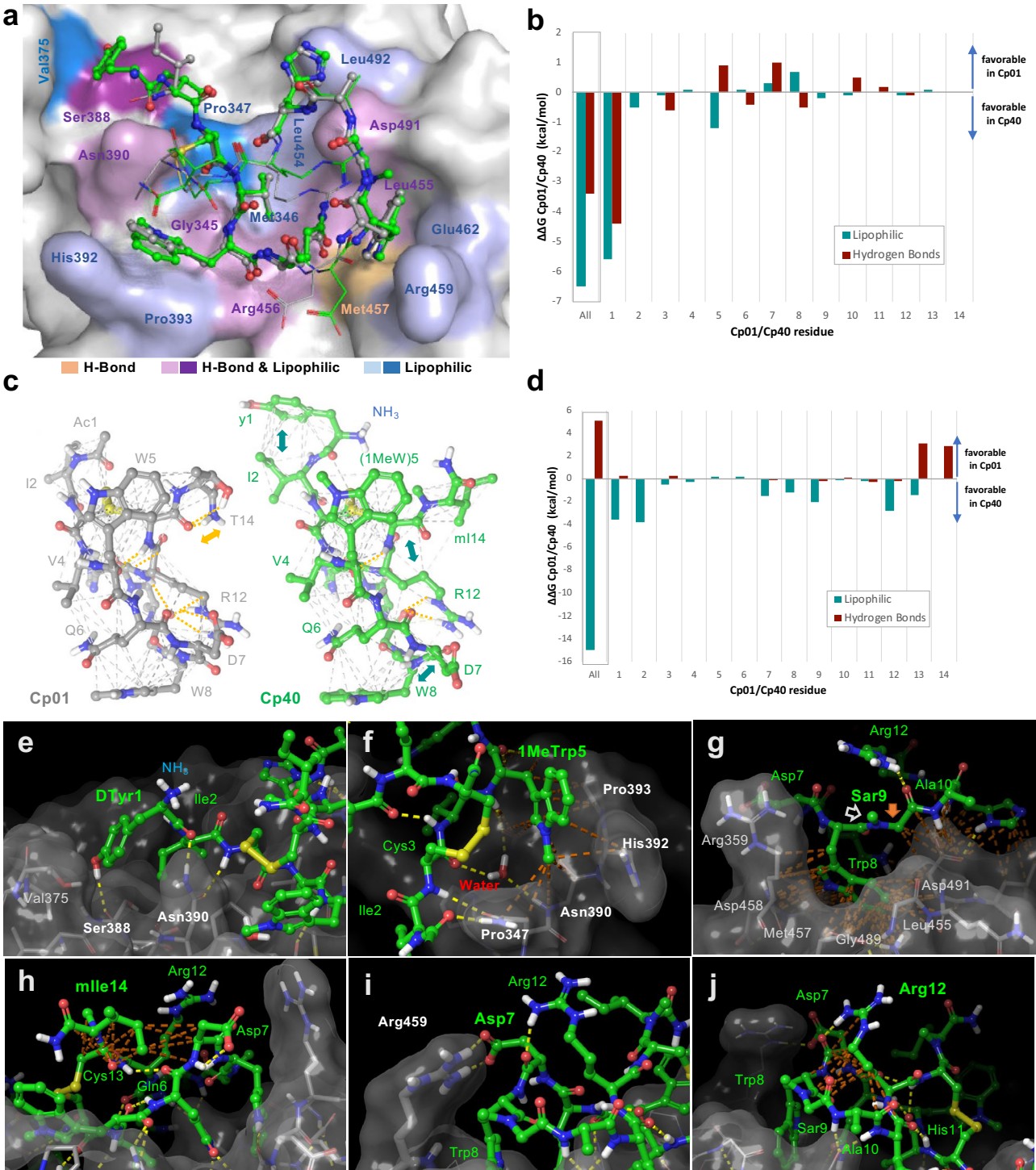

**Fig. 3 | Interaction profiles of compstatin analogs Cp01 and Cp40 showing contributions of intermolecular (a,b) and (c,d) intramolecular H-bonds and hydrophobic contacts. a** Contact residues on the target protein engaging in H-bonds (yellow), lipophilic contacts (blue) or both (purple) are highlighted on the surface of C3b. Contacts only present in Cp01 but not Cp40 are show in pastel shades. The structures of Cp01 and Cp40 are shown in grey and green ball-and-stick representation, respectively; residues not engaging in target binding are shown as lines. **c** Intramolecular contact networks in Cp01 (left) and Cp40 (right) showing H-bonds in yellow and hydrophobic contacts in grey. Analog-specific H-bonds and hydrophobic contacts are marked yellow and cyan arrows, respectively. **b, d** Differences in binding free energies per residue between Cp01-C3c and Cp40-C3b based on three independent MD simulation runs with distinct initiation parameters. (Supplementary Tables 2–5). Closeup view of the interaction profiles of **e** D-Tyr1, **f** (1Me)Trp5, **g** Sar9, **h** mIIe14, **i** Asp7, and **j** Arg12. C3b is shown in stick representation with grey semitransparent surface and Cp40 in green ball-and-stick representation. H-bonds and hydrophobic contacts are marked as yellow and brown dotted lines, respectively.

lipophilic contacts for the interactions of Cp40 with C3b (Fig. 3a, b) and the intramolecular stabilization of the bound peptide (Fig. 3c, d). We performed the same analysis for Cp01 for a direct comparison of residue-specific contributions (Fig. 3 and Supplementary Tables 2–5). Overall, Cp40 shows improved binding energy relative to Cp01 ($-103.7 \pm 7.4$ vs. $-99.8 \pm 6.6$ kcal/mol), which we attributed to a larger number of lipophilic interactions with C3b/C3c and stronger intramolecular lipophilic interactions that stabilize the binding conformation. Conversely, the replacement of the polar Thr by an apolar mIle at the C-terminus abrogated one H-bond.

The most significant contribution to the improved lipophilic contact with the target protein is mediated by the N-terminal extension of the peptide by D-Tyr1 (Fig. 3e). This contact accounts for $-6.0$ kcal/mol, which is an improvement of $-5.4$ kcal/mol compared to the acetyl group of Cp01 (Fig. 3 and Supplementary Table 2). D-Tyr1 forms H-bonds with Glu372 or Ser388 in MD simulations or the crystal structure, respectively, and accepts an H-bond from Asn390 in C3b. Furthermore, D-Tyr1 is important for stabilizing the N-terminal part of the peptide by forming intramolecular lipophilic interactions to Ile2.

During lead optimization, indole N-methylation of Trp5 strongly affected target affinity, which could not be explained by an increase in overall hydrophobicity[10,25]. Indeed, whereas the crystal structure suggests a distinct lipophilic cavity to be filled by the methyl group, MD simulations show only a slight improvement in interaction energy (1.2 kcal/mol) by the additional lipophilic interaction and the loss of H-bond formation with Met457 even has an unfavorable impact (+0.9 kcal/mol). Rather, the improved affinity can be attributed to a strong interaction of structural water fixed by Thr391 and Asn390, which is shielded by the 1-methyl group of (1Me)Trp5 (Fig. 3f and Supplementary Fig. 4). When compared to Cp01, this results in a distinct interaction pattern in which the N-terminus of Cp40 forms a strong, cooperative H-bond network with Asn390 (i.e., accepting an H-bond from the backbone NH of Cys3, donating an H-bond to the carbonyl of D-Tyr1). This interplay also stabilizes a Cp40 conformation where Ile2 engages in a favorable hydrophobic interaction with Pro347 and Val375 of C3b.

Another conformational improvement was achieved by N-methylation of Gly9 to Sar9. In contrast to the Cα atom, the added N-methyl group does not form direct hydrophobic contacts with C3b (Fig. 3g). Rather, the enhanced intramolecular stabilization determined by MD simulations (Supplementary Table 5) can be explained by steric effects that promote a bioactive conformation and orient the side chain of Trp8 to engage in favorable hydrophobic interactions with C3b (Fig. 3g). Similarly, exchanging the C-terminal Thr for mIle yielded a conformation that supports intramolecular contacts of the mIle14 N-methyl moiety with Asp7 and Arg12 (Fig. 3h). The increased contact surface of mIle supports hydrophobic packing of the C-terminal amino acids to the core of the bound macrocyclic peptide, therefore shielding the side chains from the solvent. This form of intramolecular conformational stabilization can be detected in the majority of MD simulation frames, in contrast to the solvent-exposed binding mode seen in the crystal structure. This discrepancy can be explained by the crystal packing of the C3b-Cp40 complex as visualization of the crystal mates reveals that mIle14 interacts with Met968 of a neighboring mate protein (Supplementary Fig. 5). Moreover, the side chain orientations of charged residues Asp7 and Arg12 deviated between the MD simulation and the crystal structure. Based on averaged contributions from MD simulations, the aliphatic portion of Asp7 (Cα, Cβ) contributes to the lipophilic interaction toward the C-terminal mIle (Supplementary Table 5 and Fig. 3h, i). While not defined in the crystal structure, a majority of MD frames showed an intramolecular electrostatic interaction of Asp7 with Arg459 of C3b (Fig. 3i, Supplementary Fig. 6). Arg12 is heavily involved in both lipophilic and hydrophilic intramolecular interactions. The mobility of the side chain atoms explains the poorly defined electron density in the crystal

structure and suggests dynamic interactions. In particular, hydrophobic contacts with the N-methyl of Sar, aliphatic atoms of Asp7, and the side chain of mIle14 can be observed (Fig. 3j). Additionally, the guanidinium N-H donors alternate in H-bonding with carbonyls of Sar9 and Asp7, oriented away from protein and inwards to the macrocycle. Despite forming no contacts to C3b in the co-crystal or MD simulations, Arg12 emerges as the fourth-most important residue in Cp40 when all partial interaction energies are summed.

## SAR studies rationalize previous lead optimization steps

During the past 15 years, the efficacy of the compstatin family has been iteratively improved. However, the impact of individual optimization steps on the 300-fold affinity enhancement from Cp01 to Cp40 (Fig. 1c) have not been investigated, and the role of the charged residues Asp7 and Arg12 remains unclear. To experimentally confirm computational analyses, we performed a comprehensive SAR study, in which individual amino acids in Cp40 were either reverted to the Cp01 state or substituted by residues of distinct functionality (Fig. 4a, b). We synthesized a panel of Cp40 derivatives (Supplementary Table 6) and evaluated each compound for binding affinity and kinetics using surface plasmon resonance (SPR) (Fig. 4c–e and Table 1)[14]. The interaction profile we measured for Cp40 was comparable to reported values, with subnanomolar target affinity ($K_D$), rapid association ($k_a$), and slow dissociation rate constants ($k_d$) (Table 1)[14].

The crystal structure of Cp40 with C3b shows that D-Tyr is able to address an extended, largely hydrophobic pocket and form additional contacts. Thus, we investigated the influence of the stereochemistry and side chain functionality of D-Tyr1 and synthesized derivatives featuring L-Tyr, D-Ala, and L-Ala at this position. We included analog Cp20, which corresponds to a reversion of D-Tyr with the acetylated N-terminus found in Cp01. A substitution of the residue from D-Tyr to D-Ala resulted in a faster dissociation rate with no effect on association kinetics, which yielded a 3.6-fold reduction in affinity. Similarly, a change in stereochemistry from D-Tyr to L-Tyr had little impact on association rates but accelerated dissociation rates, resulting in a 3-fold affinity drop. The corresponding conversion from D-Ala to L-Ala did not negatively impact affinity. Finally, reverting D-Tyr to the acetyl moiety of Cp01 resulted in the most profound acceleration of the dissociation rate (Fig. 4c and Table 1). These findings suggest that the affinity gain is mediated by the size and orientation of the side chain. As expected, the stereochemistry effect is more pronounced for the larger Tyr. This agrees with the MD simulations, where the phenolic OH forms H-bonds to the protein and the D-configuration aids in intramolecular stabilization of the N-terminus.

Our investigation on the influence of methylations on affinity confirmed the importance of position 5. The affinity and kinetic profile were highly diminished when reverting (1Me)Trp5 to Trp, which resulted in the least active derivative in the singly-mutated Cp40 series with a -24-fold loss in affinity (Fig. 4c and Table 1). Initial structural analysis based on the Cp01-C3c structure suggested that the methyl group faces a lipophilic groove formed by His392 and the backbone of Thr391 (Supplementary Fig. 7a, b) and that the quality of the hydrophobic interaction may be further exploited by extension of the N-substituent. Therefore, we investigated the effect of 1-indole substitution with bulkier alkyl substituents (i.e., ethyl, butyl, 3-methylbutyl). However, these derivatives had an unfavorable effect on affinity with a loss that was primarily driven by a decrease in $k_a$, whereas $k_d$ remained comparatively stable (Supplementary Fig. 7c). This indicates that larger alkyl substituents impede target binding and do not improve the interaction of Trp5 through hydrophobic contacts. Rather, as shown by MD analysis, the unique affinity enhancement attributed to the methyl substituent is caused by shielding of a buried structural water that improves cooperative H-bonding of Cp40 to Asn390. Compared to (1Me)Trp5, the influence of N-methylation in position 9 on overall affinity is relatively small (Fig. 4c and Table 1). The

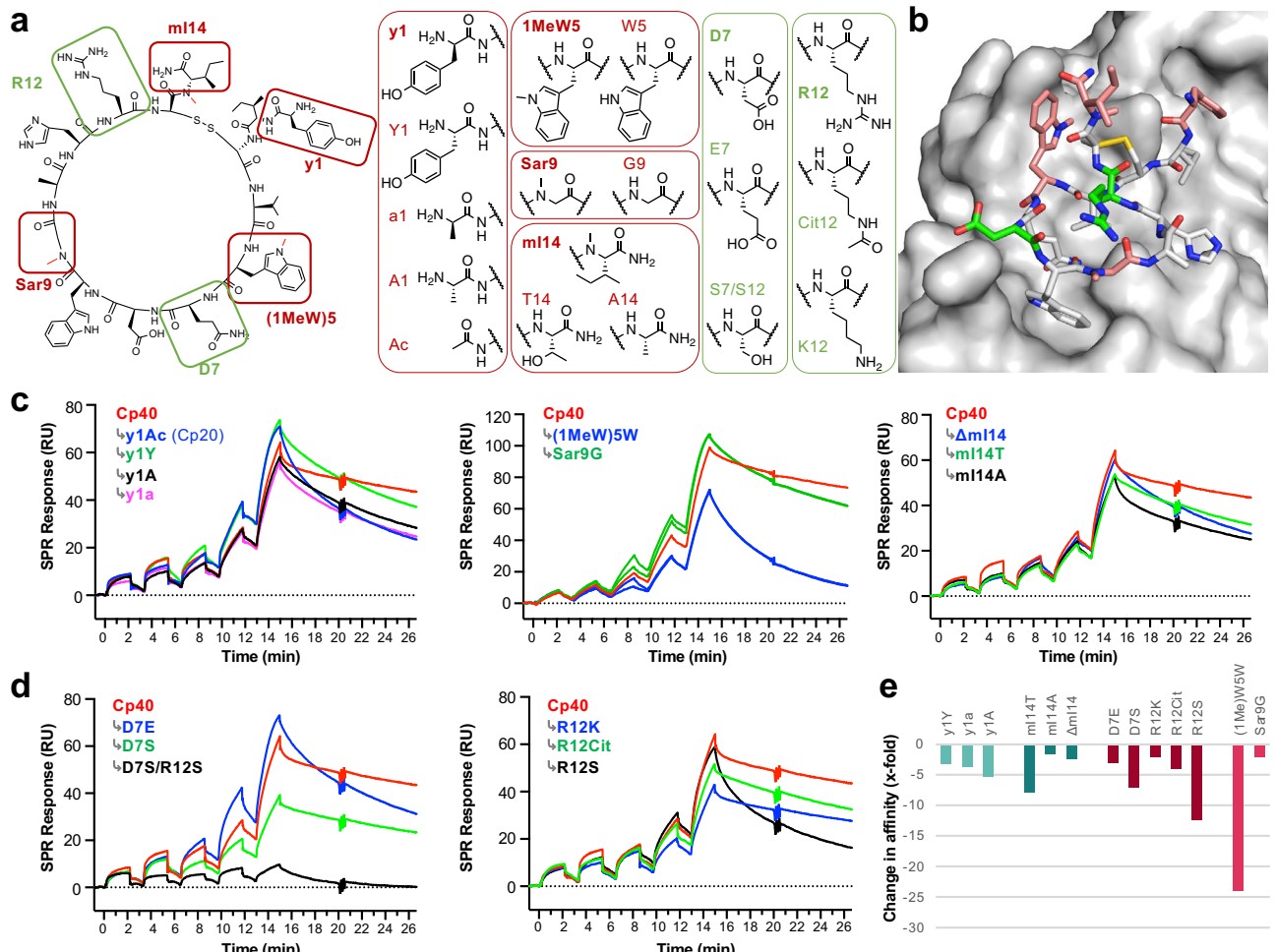

**Fig. 4 | Structure-activity relationship study of the Cp40-C3b interaction.**
**a** Amino acid substitutions introduced for the study, with residues altered during lead optimization steps framed in red and charged residues in green. **b** Localization of the changed residues on the crystal structure of Cp40 with C3b color coded as in **a**. SPR sensorgrams of Cp40 and its site-specifically mutated derivatives affecting the **c** optimized or **d** charged residues. Representative sensorgrams from at least three independent SPR experiments with comparable outcome. Average values and standard deviations are shown in Table 1. **e** Quantitative comparison of changes in binding affinities based on the results in Table 1. The overall affinity difference between Cp01 and Cp40 (i.e., 300-fold) is not shown due to scaling ranges. RU resonance units.

Sar9Gly mutant had a 2-fold drop in $k_d$ while $k_a$ remained largely unaffected. Whereas previous studies suggested that introducing Sar to obtain Cp10 had a beneficial effect on $k_a$ by facilitating a bound-like solution conformation[14], this effect is likely masked in Cp40 by the improvement in $k_a$ afforded by D-Tyr1.

The solvent-exposed orientation of the C-terminal mIle14 of Cp40 appears unlikely due to the hydrophobic nature of the side chain but may rather be attributed to crystal mate contacts (Supplementary Fig. 5). To confirm this hypothesis, we tested derivatives where mIle was replaced by Thr (as in Cp01), Ala, or removed completely to truncate the C-terminus. SPR analysis revealed a negative impact on $k_a$ and $k_d$ for the mIle-to-Thr mutant, which led to a ~8-fold loss in affinity. Both the mutation to Ala and the C-terminal truncation profoundly reduced $k_d$ with little alteration to $k_a$ (Fig. 4c and Table 1). These data confirm the contribution of the C-terminal residue to target residence, which is supported by the MD simulation that highlighted the role of mIle14 in intramolecular hydrophobic stabilization of the Cp40 bound conformation.

## Charged residues in Cp40 play important roles

In contrast to the hydrophobic mIle14, solvent exposure of the charged residues Asp7 and Arg12 is considered plausible. In the Cp01-C3c and Cp40-C3b structures, the side chains of both amino acids are resolved in conformations that face the solvent and have limited or no contact with the target protein. While this may suggest a contribution to solubility rather than affinity, the possibility of intramolecular effects has not been investigated. Indeed, our MD simulations indicate transient target interactions for Asp7. To assess potential contributions to binding, we substituted Asp7 and Arg12 with residues of the same charge or with the polar, non-charged Ser. SPR analysis revealed that Asp7 and Arg12 both have distinct impacts on binding kinetics (Fig. 4d and Table 1). The negative charge in Asp7 appears to contribute to complex formation, as $k_a$ remains unaffected by replacement with equally charged Glu, but yields a 10-fold drop when the neutral Ser is introduced. Such charge-dependent effects on association kinetics may be attributed to long-range electrostatic steering. The impact of position 7 on target residence (i.e., $k_d$), with the side chain extension in Glu7 decreasing (3-fold) and a replacement by Ser actually improving (1.4-fold) complex stability, were unexpected. The 7-fold affinity loss of the Asp7Ser mutant can be largely attributed to the aforementioned 10-fold drop in $k_a$. For Arg12, replacement of the guanidinium group by a primary amine in Arg12Lys yielded a 2-fold slower $k_a$. This may be primarily mediated by charge and H-bonding rather than side chain length, since a citrulline derivative behaves similarly to Lys. Of note, $k_d$ remained unaffected among all positively charged derivatives, but an exchange with Ser led to a marked loss of

**Table 1 | Mutations and corresponding affinities and kinetic profiles evaluated in the SAR study**

| Position[a] | Compound[b] | $K_D$ [nM] | $k_a$ [$10^6$ M$^{-1}$ s$^{-1}$] | $k_d$ [$10^{-3}$ s$^{-1}$] |
|---|---|---|---|---|
| | Cp40 | $0.8 \pm 0.2$ | $1.0 \pm 0.4$ | $0.7 \pm 0.1$ |
| | Cp40[c] | $0.5 \pm 0.1$ | $2.8 \pm 0.5$ | $1.4 \pm 0.1$ |
| | Cp20 | $3.4 \pm 0.6$ | $1.3 \pm 0.3$ | $4.3 \pm 1.3$ |
| 1 | Cp40 y1Y | $2.5 \pm 0.9$ | $0.6 \pm 0.3$ | $1.3 \pm 0.1$ |
| | Cp40 y1A | $3.8 \pm 0.5$ | $0.4 \pm 0.0$ | $1.5 \pm 0.4$ |
| | Cp40 y1a | $2.9 \pm 0.6$ | $0.8 \pm 0.1$ | $2.3 \pm 0.4$ |
| 5 | Cp40 (1MeW)5 W | $19.2 \pm 2.0$ | $0.2 \pm 0.0$ | $2.7 \pm 0.7$ |
| 9 | Cp40 Sar9G | $1.7 \pm 0.7$ | $0.9 \pm 0.4$ | $1.2 \pm 0.2$ |
| 7 | Cp40 D7E | $2.4 \pm 0.5$ | $0.9 \pm 0.2$ | $1.9 \pm 0.1$ |
| | Cp40 D7S | $5.6 \pm 1.3$ | $0.1 \pm 0.0$ | $0.5 \pm 0.2$ |
| 12 | Cp40 R12Cit | $3.2 \pm 0.6$ | $0.6 \pm 0.1$ | $1.9 \pm 0.0$ |
| | Cp40 R12K | $1.6 \pm 0.2$ | $0.5 \pm 0.1$ | $0.8 \pm 0.0$ |
| | Cp40 R12S | $9.9 \pm 0.5$ | $0.2 \pm 0.0$ | $1.8 \pm 0.3$ |
| 7/12 | Cp40 D7S/R12S | $59.1 \pm 41.3$ | $0.1 \pm 0.0$ | $4.0 \pm 1.0$ |
| 14 | Cp40 mI14T | $6.2 \pm 0.8$ | $0.2 \pm 0.0$ | $1.4 \pm 0.1$ |
| | Cp40 mI14A | $1.2 \pm 0.6$ | $3.0 \pm 1.6$ | $3.0 \pm 0.4$ |
| | Cp40 ΔmI14 | $1.8 \pm 0.2$ | $1.5 \pm 0.2$ | $2.7 \pm 0.2$ |

[a]Residue position based on the Cp40 numbering, [b]abbreviations used: *1MeW* 1-methyl tryptophan, *Sar* sarcosine, [c]reported values from Qu et al.[14]. Values represent average±standard deviation from three independent SPR experiments on amine-reactive immobilized C3b.

affinity (-12-fold) with impacts on both $k_a$ and $k_d$ (Table 1). Apparently, Ser cannot mimic the complex intramolecular stabilization signature of Arg. The affinity loss is even stronger when both charged amino acids are replaced with Ser due to a faster $k_d$ (Fig. 4d). Overall, and in line with MD simulations, this suggests primary roles of Asp7 and Arg12 in both electrostatic steering and intramolecular stabilization due to lipophilic interactions, although Asp7 may also form a transient salt bridge to C3b (Supplementary Fig. 6).

## Cp40 acts on C3 substrate activation rather than convertase formation

It is now established that convertase formation in the amplification loop follows a tiered process, in which FB binds to C3b and undergoes a conformational change that allows FD to bind and enzymatically remove the Ba segment. The Bb segments remains bound to the metal ion-dependent adhesion (MIDAS) site at the C-terminal complement (CTC) domain of C3b, thereby forming the C3 convertase complex (i.e., C3bBb; Fig. 5a)[5]. Upon binding of C3, the flexible attachment of the CTC domain enables Bb to move proximal to the scissile bond at the ANA domain of C3. This results in the removal of C3a and a drastic conformational rearrangement of C3b that exposes the highly reactive acyl imidazole moiety, which allows its covalent deposition at the immediate site of activation (Fig. 5a).

The MG4/5 domain region of the C3b β-chain, which harbors the compstatin binding site, does not seem to be involved in convertase formation but rather C3 substrate binding. Previous studies have shown that two C3b molecules can interact via a dimerization interface involving MG4/5, which is also present in C3[6]. This finding suggests that C3 binds to the convertase via this dimerization interface, and that compstatin analogs may therefore interfere with the initial binding of C3 to C3bBb[17]. Indeed, superimposition of the crystal structure of C3b-Cp40 obtained in this study with the structures of C3 and SCIN-stabilized dimeric C3bBb confirmed that the interactions of compstatin with the C3 substrate and convertase C3b lead to steric hindrance (Fig. 5b, Supplementary Fig. 8). However, these structure-based predictions still need experimental confirmation.

To investigate whether Cp40 interferes with convertase formation and/or C3 activation, we performed an SPR assay to monitor convertase

mechanisms (Fig. 5c–e). When a mixture of FB and FD was injected to physiologically oriented C3b on a sensor chip, we observed the initial formation of the C3bBb convertase followed by a steady decay. Subsequent injection of C3 during the decay phase initiated a substantial signal increase from the covalent deposition of C3b on the hydroxyl-rich sensor chip matrix (Fig. 5c, blue curve). Injection of C3 in the absence of convertase formation did not result in any increase and showed a weak binding signal (Fig. 5c and grey curve). We repeated this assay with addition of Cp40 to the running and sample buffers. Convertase formation or decay was not impaired in the presence of Cp40. The elevated C3bBb signal observed in presence of Cp40 can be attributed to C3b deposition from the previous step. Importantly, Cp40 completely abrogated C3b deposition upon C3 injection (Fig. 5c, red curve), which confirmed that compstatin analogs interfere with C3 activation rather than convertase formation or stability, as natural regulators do.

Surprised by the weak-affinity nature of the physiologically important interaction between C3 and surface-immobilized C3b, we then investigated whether binding of C3 to the intact convertase complex is distinct from free C3b (Fig. 5d). To circumvent rapid convertase decay and prevent a signal overlay by C3b deposition, we used a mutant of FB, which contained a Ser-to-Ala mutation in the catalytic site to render it inactive[5]. Injection of C3 on a C3b chip yielded negligible binding, confirming the weak interaction. In the presence of a stabilized inactive C3bBb convertase, however, subsequent C3 injection caused very strong binding. In contrast to the experiment with an active convertase (Fig. 5c), the C3 signal showed a rapid decay (Fig. 5d), which reflected reversible C3 substrate binding without activation and covalent deposition of C3b. We then queried the influence of Cp40 on the binding of C3 to the stable, inactive convertase. In alignment with the results obtained with the active convertase (Fig. 5c), injection of C3 to the preformed convertase induced a binding signal, whereas no binding event occurred in the presence of Cp40 (Fig. 5e). We also evaluated whether the binding of Cp40 to the substrate (C3) or the convertase (C3bBb) primarily contributes to the protein-protein interaction inhibition that prevents C3 activation. Our SPR results indicate that the blockage of either side has a profound impact on C3 binding to the convertase with an increased efficacy if both target sites are occupied by Cp40 (Fig. 5f, g).

In addition to the C3-C3b dimerization, interactions between C3 and C4b and between C5 and C3b and/or C4b are considered critical for complement activation processes. Whereas previous studies established that Cp40 inhibits the C3-C4b interaction only partially[26], we now also investigated the binding of C5 to the C3bBb complex and observed partial inhibition of the interaction in saturating concentrations of Cp40 (Fig. 5h). As expected, due to the lack of binding sites, Cp40 did not influence the binding of C5 to C4b (Supplementary Fig. 9).

Overall, our experimental studies corroborate the structure-derived hypothesis that compstatin analogs prevent the initial binding of the C3 substrate to C3bBb. Without this step, C3 cleavage and C3b deposition cannot take place, thereby impairing opsonization, amplification, and effector generation via the amplification loop.

## Mono- and bivalent compstatin analogs exert distinct target selectivity modes

Owing to the common binding site, compstatin analogs can bind to circulating C3, the surface-tethered opsonin C3b, and the convertase complex C3bBb (Fig. 1a). An interaction with the C3b breakdown fragments iC3b (surface) and C3c (circulation), which do not participate in amplification, is also possible. We investigated whether binding of compstatin analogs to circulating and surface-bound targets is equivalent, and whether the binding mode may be influenced by the distinct molecular configurations of pegcetacoplan and AMY-101 (Fig. 6a). In the case of Cp01, the binding affinities for soluble C3c and immobilized C3b were highly comparable when determined by isothermal titration calorimetry (ITC) and SPR, respectively (Fig. 6b and

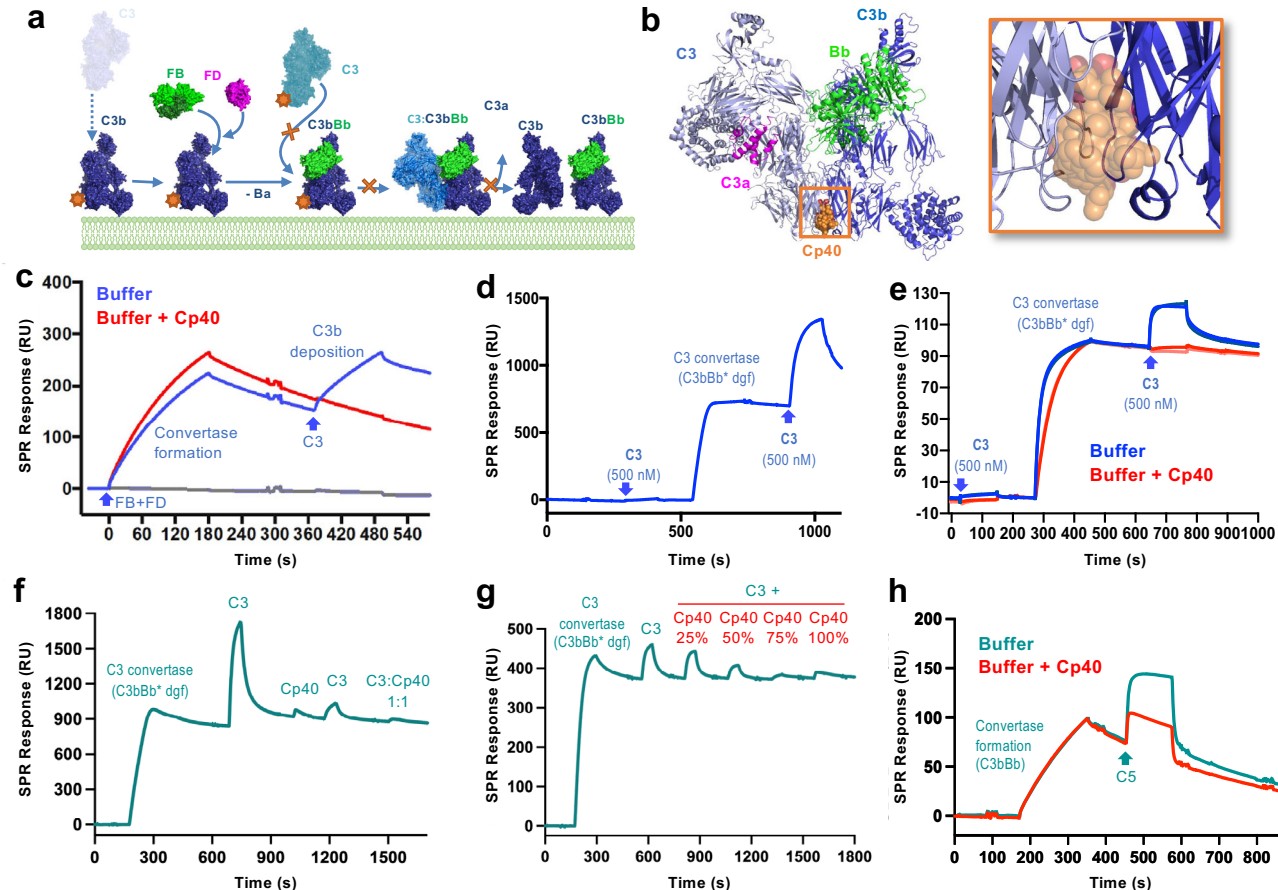

**Fig. 5 | Target binding and functional mechanism of compstatin analogs.**
**a** Structure-based hypothesis of C3 activation, opsonization, and amplification using the same structures as in Fig. 1b. Target binding and inhibition by compstatin analogs is depicted as orange stars and crosses, respectively. **b** Superimposition of C3b-Cp40 and C3-Cp40 on the structure of a dimeric, stabilized C3 convertase, which refines the model of C3 inhibition by suggesting that Cp40 sterically hinders the binding of C3 to C3bBb. **c** SPR analysis of the influence of Cp40 on convertase formation and activity. The presence of Cp40 (red) does not inhibit formation or decay of C3bBb, but prevents subsequent C3b deposition upon C3 injection in absence of the inhibitor (blue), similar to the complete absence of a convertase (grey). **d** Selectivity of soluble C3 for C3bBb (formed in a stabilized, inactive version) over free C3b on a sensor chip. **e** Cp40 prevents C3 binding to C3bBb. Upon formation of the inactive convertase, notable binding of C3 was observed in the absence but not presence of Cp40, thereby confirming that compstatin analogs impair the initial C3 substrate binding step. **f** The binding of Cp40 to the C3bBb complex leads to a marked yet incomplete impairment of substrate binding (i.e., free C3), whereas the presence of Cp40 on C3 and C3bBb causes full inhibition. **g** The selective Cp40-mediated inhibition of the C3 substrate is concentration dependent. **h** C5 interacts with the C3 convertase (C3bBb) (cyan), with the presence of Cp40 (red) leading to a partial reduction of the interaction. **c–h** Representative sensorgrams of at least 2 independent SPR experiments with comparable outcome. dgf double gain-of-function mutant, FB factor B, FD factor D, RU resonance units.

Supplementary Fig. 10a). Although subsequent affinity enhancements increasingly limit a reliable quantification by ITC, a correlation between solution and surface binding can also be observed for Cp05 and Cp40 (Supplementary Fig. 10a). All three analogs show a 1:1 binding mode between target and inhibitor in ITC (Fig. 6b). In contrast, the conjugation of two Cp05 entities to a central 40-kDa PEG moiety in analogy to pegcetacoplan results in an At/Mt ratio of 1/0.5, supporting a mode in which one inhibitor entity can simultaneously bind two target proteins in solution (Fig. 6a, b). When compared to its monovalent precursor, Cp05-PEG-Cp05 showed a notable drop in target affinity, which may be related to diffusion and target accessibility effects. Whereas the bivalent presentation of Cp05 may have comparatively little impact in solution, the linkage of active entities may become more relevant for the binding to surface-bound C3b. Indeed, the binding of Cp05-PEG-Cp05 to immobilized C3b on an SPR sensor chip was profoundly enhanced when compared to monovalent Cp05 (Fig. 6c, d). This effect was mediated by a strongly reduced dissociation rate, likely due to a bridging of C3b molecules. Of note, both the association rate and the binding capacity of Cp05-PEG-Cp05 was notably decreased, which again may indicate an impact of PEG on target accessibility (Fig. 6c, d and Supplementary Fig. 10b). Our studies indicate that bivalent

compstatin analogs show a distinct activity mode with a preference for surface-bound C3b, whereas monovalent analogs bind equally well to circulating and surface-deposited C3-based targets.

## Missing contacts rather than steric hindrance define the species specificity of Cp40

Though compstatin analogs bind to a conformationally stable and largely conserved region on the β-chain of C3, initial studies with compstatin showed that the peptide inhibited the human and not the murine complement pathway[8]. Compstatin was shown to have narrow species specificity for human and non-human primate C3 (NHP; e.g., baboons and cynomolgus monkeys) with no inhibitory activity confirmed for murine, guinea pig, rabbit, or porcine serum[27]. Since the drastic improvement of C3 binding affinity by compstatin analogs, we anticipated that some gain of activity of the sub-nanomolar Cp40 may be conceivable for other species. We conducted complement inhibition assays based on antibody-mediated activation in human, mice, rat, rabbit, dog, and pig plasma. As expected, strong inhibitory activity was observed in human plasma, whereas no notable inhibition of complement activation by Cp40 was observed in non-primate species (Fig. 7a). Despite a profound increase in target binding for human C3,

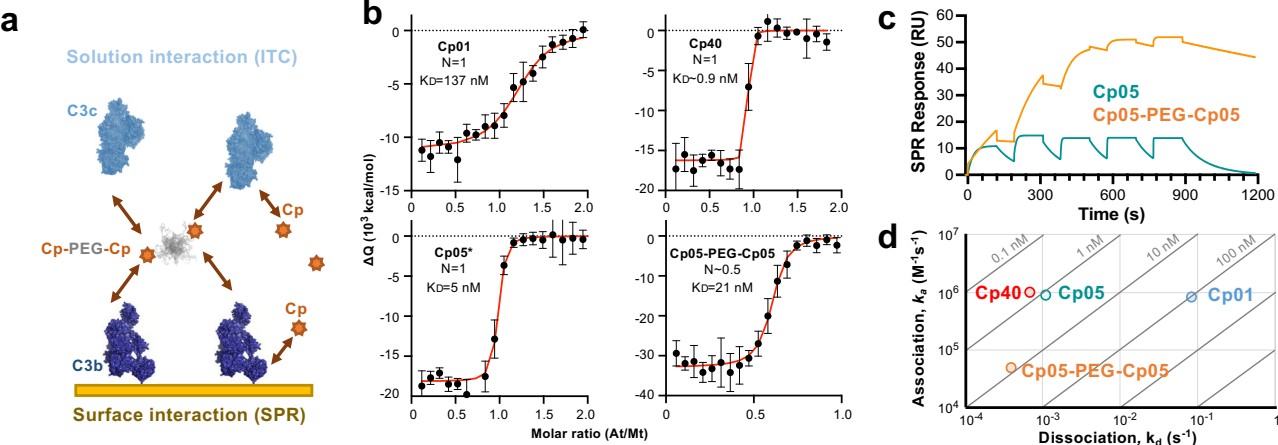

**Fig. 6 | Differential interaction modes of mono- and bivalent compstatin analogs. a** Compstatin-based drugs may interact with circulating (i.e., C3, C3c) or surface-tethered targets (i.e., C3b, C3bBb, iC3b). Depending on their valency, avidity may occur. **b** ITC analysis of solution interactions between C3c and compstatin analogs Cp01, Cp05*, Cp05-PEG-Cp05, and Cp40 shows 1:1 binding for monovalent analogs ($N = 1$) whereas the data of the bivalent Cp05-PEG-Cp05 fits a 2:1 ratio ($N = 0.5$). The displayed curve is a representative of two independent experiments per compound, obtained by peak-to-peak integration of the processed thermogram, as a function of At/Mt (titrant-to-titrate molar ratio). The integrated signal (black) is fitted to a 1:1 binding model (red curve) for determination of affinity ($K_D$) and molecular ratio ($N$). Error bars represent the uncertainty associated to the integral calculation of each peak in the binding isotherm (as determined by AFFINImeter). Cp05* contains the linker necessary for the conjugation to PEG-40kDa; its binding profile is comparable to that of Cp05 (Supplementary Fig. 10c). **c** SPR analysis between immobilized C3b and mono- and bivalent Cp05 analogs reveal a strong enhancement of surface binding for Cp05-PEG-Cp05 over monovalent Cp05. **d** Isoaffinity kinetic plot for surface interactions of compstatin analogs with immobilized C3b, demonstrating that the development steps from Cp01 to Cp40 were enabled by improving dissociation rates with little impact on association rates. In contrast, the improved stability of the C3b complex with Cp05-PEG-Cp05 is accompanied by a marked drop in the association rate. PEG polyethylene glycol, RU resonance units.

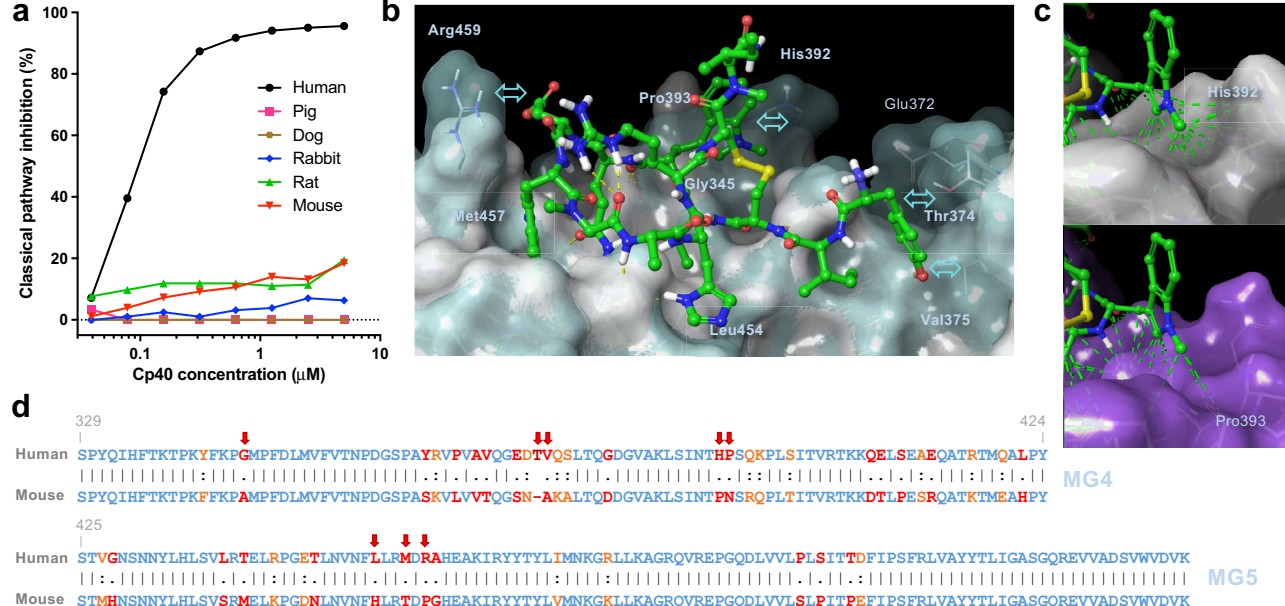

**Fig. 7 | Species specificity of Cp40 for human and non-human primate complement. a** Complement-inhibitory activity of Cp40 in human, pig, dog, rabbit, rat, and mouse plasma as assessed by classical pathway ELISA. Negative values have been defined a zero inhibition. Raw data and calculated values are available in the source data file. **b** Comparison of Cp40 binding to human C3b (based on crystal structure, shown as transparent and light blue surface) and mouse C3b (homology model, shown as grew surface). Labels correspond to human C3. **c** Close-up view on the species specificity of current compstatin analogs remains limited to humans and non-human primates. (1Me)Trp5 and its interaction to C3b. In the case of mouse C3b, relevant interactions to His392 are missing. **d** Pairwise sequence alignment (EMBOSS Needle) of human and mouse C3 within the MG4/MG5 domain region. Identical, strongly similar, and weakly similar residues are highlighted in blue, orange, and red, respectively. C3 residues engaging in intermolecular interactions with Cp40 are marked with arrows.

We elucidated the molecular determinants of this narrow species specificity (Fig. 7b, c) from our crystal structure of Cp40 in complex with human C3b and a homology model of the β-chain of mouse C3.

The mouse C3b model showed a structurally consistent and similar fold to the human C3b-Cp40 structure (Supplementary Fig. 11a). Cp40 bound to mouse C3b assumed an identical conformation to human C3b-Cp40 (RMSD = 0.63 Å). No backbone clashes with murine C3b were visible and the intra- and inter-molecular H-bonding patterns

were consistent with human C3b-bound Cp40. This suggests that species differences in binding likely stem from altered steric, hydrophobic, and electrostatic interactions. Comparative surface mapping of the human and mouse C3b structures identified three structurally distinct areas within the Cp40 binding site (Fig. 7b and Supplementary Fig. 11b). Firstly, the loop portion defined by His392 and Pro393 in human C3b is differently constituted in mouse C3b (Pro392, Asn393) and lacks the structured cavity to properly accommodate (1Me)Trp5 (Fig. 7b, c). Secondly, the charged residue Arg459 is replaced by a lipophilic Pro in mouse C3b. While the missing positive charge may negatively affect long-range electrostatic interactions, Asp7 may also lose the transient salt bridge to Arg459 in human C3b as suggested by MD simulations. Finally, the extended binding site defined by residues 372–377 in human C3b, which engages with D-Tyr1 of Cp40, is differently constituted in mouse C3b and is shifted away from Cp40 (Fig. 7b), preventing effective stabilization of the bound ligand. These analyses suggests that the species specificity of Cp40 is defined by a loss of key interaction partners that prevent ligand binding.

## Discussion

The compstatin family of C3 inhibitors is a promising drug class for the treatment of various complement-mediated diseases. Despite the continued development of compstatin analogs, several aspects about their binding profiles and mode-of-action remained elusive. We have solved the crystal structure of the clinical candidate Cp40 with one of its natural targets, C3b, and combined these studies with multi-level analyses, to decipher the binding mode of Cp40 in great detail. From these data, we provide molecular-level explanations for the marked increase in target affinity and residence of compstatin analogs that was achieved during the past two decades.

Comparison of the Cp40-C3b crystal structure with the previously reported Cp01-C3c structure[17] confirmed both the target binding site at the MG4/MG5 domain interface and the square-shaped conformation of the bound peptide. SPR- and MD-based SAR studies revealed that complex intramolecular peptide interactions and optimized target contacts contribute to the subnanomolar affinity of Cp40. As free compstatin analogs are believed to fluctuate between open and closed, bound-like conformations[13,14,28], intramolecular interactions may influence target binding, induced fit, binding stability entropy, and solvent exposure. In agreement with computational studies[28], H-bonds and hydrophobic interactions largely define the contact network between C3b and Cp40. Compared to Cp01, the number and quality of target interactions are improved in Cp40, which yields the 300-fold affinity gain between the analogs. The N-terminal D-Tyr and the (1Me)Trp at position 5 were confirmed as main contributors to the enhanced interaction profile. Our study suggests the effect of (1Me)Trp can be attributed to the shielding of structural water bound by Asn390 and Thr391 and formation of a network of cooperative H-bonds rather than hydrophobicity. We have also identified an extended binding pocket on C3b that forms key contacts with the side chain of D-Tyr1. The N-methyl group of Sar9 does not engage in target interactions, but influences the orientation of adjacent residues and intramolecular contacts, though with limited relevance to overall Cp40 binding. These findings will guide future optimization and development of analogs.

We identified three residues in the Cp40-C3b crystal structure whose side chains did not participate in target interactions. While the unexpected orientation of mIle14 could be attributed to crystal packing effects, this residue indeed appears to mediate target binding via intramolecular stabilization of Cp40 rather than C-terminal interactions with C3b. Similarly, Arg12 does not form direct contacts with C3b but has an impact on intramolecular peptide stabilization. While not visible in the crystal structure, MD simulations reveal that Asp7 can form a transient salt bridge with the target, and our SAR studies suggest that both charged residues affect ligand binding via electrostatic steering. Our study thereby provide evidence that Asp7 and Arg12 not

only confer solubility in the compstatin family but also define their target interactions. Overall, insights from our characterization of the Cp40 binding profile will provide a design basis for next-generation compstatin analogs.

Moreover, our functional analyses confirm and extend previous hypotheses that compstatin analogs act as protein-protein interaction inhibitors that prevent the binding of C3 to the amplification loop C3 convertase to impair C3b-mediated opsonization and effector generation. Intriguingly, our binding studies uncovered a strong binding preference of soluble C3 for convertase-associated C3b (i.e., C3bBb) over free C3b, which suggests an active recruitment of C3 upon convertase formation to quickly drive opsonization. This finding underscores the importance of physiological convertase regulation and the impact of therapeutic interference by Cp40.

Despite compstatin being sometimes referred to as convertase inhibitor, our studies show that Cp40 does not notably affect the assembly or stability of C3 convertases. Instead, our models show that binding of Cp40 to the MG4/5 region sterically prevents an interaction via the C3:C3b dimerization interface. The resulting blockage of C3 substrate binding to the convertase has experimental support from a study of patients with a mutation in the MG4 domain of C3[29]. This p.M373T mutation was shown to impair the binding of C3 to C3b similar to the effect of Cp01 on normal C3/C3b. In those SPR experiments, a confirmation of Cp01's mode-of-action proved challenging due to the weak interaction between C3 and surface-immobilized C3b and the presence of residual C3b-binding proteins (e.g., FH, C5) in C3 purified from patient plasma[29]. The binding studies performed here clearly demonstrate that Cp40 is able to fully prevent binding of C3 to the C3bBb complex and provide a rationale for the strong inhibitory action on the amplification loop (i.e., via the alternative complement pathway).

Compstatin analogs also exert inhibitory activities for complement activation via the classical or lectin pathways, both in vivo and in vitro[30–33]. Whether the same molecular mechanisms apply to the effect of Cp40 on C4b-driven complement activation remains to be confirmed. A direct comparison of convertase activities on a molecular level is hampered by the fact that the classical pathway C3 convertase, (i.e., C4b2b, following current nomenclature recommendations)[34], cannot be reliably formed on SPR sensor surfaces. Recent studies showed that C3 interacts with surface-bound C4b and that Cp40 only partially inhibits this interaction[26]. This may indicate differences between the C3-C4b and C3-C3b dimerization interfaces or that inhibitor binding to both substrate and convertase is required for optimal inhibition as compstatin analogs only interact with C3 but not with its ortholog C4[27]. However, in analogy to C3bBb, it is possible that C3 inhibition by Cp40 may be more pronounced for the convertase (i.e., C4b2b) than with free C4b.

Similarly, a potential impact of Cp40 on the C5 convertases that are typically depicted as C3b-containing complexes (e.g., C3bBb3b) cannot be investigated in molecular detail since the exact composition of those convertases remains matter of debate. Among the current hypotheses is a mechanism, in which the interaction with dense C3b deposits on target cells would recruit C5 and potentially induce a conformational change (i.e., priming) that renders C5 susceptible to cleavage by C3 convertases[26]. Even if our studies indicate a reduction of C5 to the Cp40-saturated C3bBb complex (Fig. 5h)[26], the overall effect of Cp40 on C5-derived effectors may likely be determined by a decreased deposition of C3b rather than direct interference with the recruitment and/or activation of C5.

A C3-bypass pathway has recently been postulated in in vitro models of fulminant complement activation via the classical pathway, which conveys C5 activation without prior C3b deposition[26]. Under these circumstances, Cp40 did not prevent complement-mediated lysis due to the lack of major C3 involvement[26]. Although mechanistically relevant, it remains to be determined in which clinical

conditions classical pathway-driven C3 bypass processes define a pathological complement response. PNH, which features uncontrolled complement activation that results in hemolysis, is mediated by the alternative pathway and is highly sensitive to C3 inhibition[35]. Indeed, Cp40 was shown to have therapeutic effects in PNH[36], and pegcetacoplan has meanwhile been approved for this indication[37]. In contrast to PNH, however, most complement-mediated disorders may not require full inhibition but would benefit from partial reduction of complement activity to restore the activation-regulation balance[1].

We have shown that current compstatin analogs remain species-specific for human and NHP complement, despite, or even due to, improvements in target affinity. A comparison between our co-crystal structure and a homology model of mouse C3b suggests that compstatin analogs should fit the corresponding site on the murine target, but lack essential interaction partners. Importantly, this reduced contact network is particularly pronounced for residues that were introduced during the optimization from Cp01 to Cp40 (i.e., D-Tyr1 and (1Me)Trp5). Even if based on models, the species specificity analysis may provide opportunities to tailor future generations of compstatin analogs to broaden the selectivity for translational research in animal models. Of note, key residues on mouse C3b that define the species specificity show stronger homology or even identity to C3 from other relevant species such as rats, dogs or pigs (Supplementary Fig. 11c).

Despite sharing the target binding site and principal mode of action, monovalent and bivalent compstatin derivatives appear to exert distinct activity profiles. The elimination half-life of monovalent analogs is largely determined by their binding to C3 as abundant plasma protein (~1 mg/ml)[7]. Whereas the long target residence of Cp40 facilitates an application of the unmodified peptide, the reduced residence time of Cp05 is compensated in pegcetacoplan by the introduction of a 40-kDa PEG moiety between two Cp05 entities to reduce renal filtration[12]. In interaction analyses, all monovalent analogs (Cp01, Cp05, Cp40) showed 1:1 binding to soluble and surface-bound C3 fragments with comparable affinity. In contrast, the bivalent configuration in Cp05-PEG-Cp05, which was used as surrogate of pegcetacoplan, enables the simultaneous binding of two target proteins. Indeed, Cp05-PEG-Cp05 showed 2:1 binding in solution with slightly weaker affinity compared to the monovalent precursor, which may be attributed to steric and/or diffusion effects of the large PEG moiety. On surface-bound C3b, the bridging effect became more apparent as the bivalent analog showed an improvement in complex stability yet at lower assembly rates and capacities. Our studies therefore indicate that the activity of bivalent compstatin derivatives such as pegcetacoplan may be partially influenced by the level of preexisting or fulminantly occurring C3b deposition as e.g., seen in PNH. Whether the potential trapping of PEGylated moieties on cell surfaces may have an impact on the efficacy and/or safety remains to be determined. Cp40 generally showed stronger target affinities when compared to Cp05-PEG-Cp05, with a particular benefit in solution (factor~25) due to its low dependence on C3b deposition. This increased capacity to bind and protect the C3 substrate rather than targeting C3b opsonins may confer an additional benefit in indications that are not exclusively driven by the alternative pathway or massive C3b deposition. Finally, the 20-fold size difference between pegcetacoplan and Cp40 may also affect tissue penetration of the unbound drugs. In recent clinical trials, pegcetacoplan treatment resulted in a marked increase in plasma C3, whereas no comparable target elevation could be observed during dosing with the monovalent AMY-101[12,38]. It will be critical to elucidate the clinical relevance of the distinct target binding profiles to guide the application of different compstatin-derived drugs in the future.

Our extensive characterization of Cp40, a promising clinical candidate for complement-mediated disorders, may pave the way for the development next-generation C3 inhibitors yet also expanded the knowledge about the mechanisms of complement activation, which will have broad utility for continuing research.

## Methods

### Reagents and proteins

Rink amide MBHA resin (0.9 mmol/g, 100–200 mesh) was purchased from Novabiochem (Darmstadt, Germany). Oxyma (ethyl-2-cyano-2-[hydroxyamino]acetate) was purchased from CEM (Kamp-Lintfort, Germany). The sources of all amino acids are provided in Supplementary Methods. The following reagents were purchased from Sigma-Aldrich (Buchs, Switzerland): piperazine (ReagentPlus 99%), 1-methyl-2-pyrrolidinone (NMP, ReagentPlus 99%), triisopropylsilane (TIS, 98%), ethanedithiol, and trifluoroacetic acid (TFA, HPLC 99%). N, N'-Diisopropylcarbodiimide (DIC, 99%) was purchased from Carbolution (St. Ingbert, Germany), and the solvents dichloromethane (DCM, HPLC grade), dimethylformamide (DMF), ethanol (EtOH, absolute grade), and acetonitrile (ACN, HPLC grade) were obtained from VWR (Dietikon, Switzerland). All chemical reagents were used without further purification.

Human C3, C3b, FH, FB, and FD used in binding and functional studies were purchased from Complement Technology (Tyler, TX, USA); C3c for ITC experiments was obtained from Lee Biosolutions (Maryland Heights, MO, USA). For crystallization experiments, C3 was purified from human plasma and converted to C3b. For this purpose, C3 (30 μM) was incubated with FB (10 μM) and FD (0.3 μM) for 20 min at 37 °C in PBS pH 7.4 containing 10 mM $MgCl_2$. The C3b fragment was subsequently isolated by anion exchange (Mono Q 5/50) and size exclusion chromatography (Superdex 200 10/300)[5,17]. Cp40, synthesized by Bachem (Bubendorf, Switzerland), was used for these experiments. Plasma from different species (mouse, rat, rabbit, dog, pig) was ordered from and collected in EDTA tubes by Cocalico Biologicals (Stevens, PA).

### Crystal-structure determination

Crystals of C3b-Cp40 were grown in 6.66% w/v polyethylene glycol 8000, 66.6 mM sodium chloride, and 33.3 mM disodium phosphate/citric acid pH 4.0 using the hanging drop vapor diffusion method at 18 °C. Crystals were soaked for several minutes in reservoir solution supplemented with 25% w/v PEG 2000 and then flash frozen in liquid nitrogen. Diffraction data were collected at European Synchrotron Radiation Facility (ESRF) beamline ID29 and were processed by XDS (build 20160514) and AIMLESS (v0.3.8)[39,40]. Crystals of C3b-Cp40 complex exhibited space group P2₁ (a = 101.5 Å, b = 90.5 Å, c = 140.1 Å, α = 90.0°, β = 108.8°, γ = 90.0°) and diffracted to a resolution of 2.0 Å. The structure was determined by molecular replacement with Phaser (v2.5.6)[41] using known structures of C3b-FH CCP1–4 (PDB 2WII)[21]. The Cp40 molecule was manually built in Coot (v0.8)[42]. The complex structure was further refined in Phenix (v1.18.2)[43], yielding R-work of 0.191 and R-free of 0.223 with acceptable stereochemistry (see Table 2 for crystallographic summary).

### Computational analysis of Cp40-C3b complex

Protein-ligand complexes for MD simulations were prepared as follows: 1) the crystal structure of C3b with Cp40 was prepared using the Protein Preparation Wizard as implemented in Maestro (Schrodinger Inc.). Missing residues 76–77 were reconstructed and hydrogen atoms were added, assuming the physiological pH of 7.4. After protein-protein alignment, the binding site waters resolved at chains D and H of the C3c-compstatin complex (PDB 2QKI) were added to the model. Polar hydrogens were oriented to form the most reasonable H-bonding network. 2) The C3b-Cp01 structure was prepared by copying Cp01 (termed 4W9A in the PDB) from PDB 2QKI to the current C3b-Cp40 crystal structure, analogous to the procedure with binding site waters.

The MD simulations in pure water were conducted using the Desmond simulation engine (v.2019-1)[44]. Using the System Builder, the

**Table 2 | Crystallographic data, including experimental conditions, unit cell dimensions, data collection, and refinement conditions**

| Dataset | C3b-Cp40 |
|---|---|
| Wavelength (Å) | 0.9677 |
| Resolution range[a] (Å) | 47.8–2.0 (2.07–2.0) |
| Space group | P 1 21 1 |
| Unit cell (a, b, c, α, β, γ) | 101.5 90.5 140.1 90 108.8 90 |
| Unique reflections | 159,011 (15,635) |
| Completeness (%) | 98.1 (97.0) |
| Mean I/sigma(I) | 9.4 (0.94) |
| Wilson B-factor (Å$^2$) | 38.8 |
| R-merge | 0.094 |
| R-meas | 0.108 |
| R-pim | 0.052 |
| R-work | 0.191 (0.355) |
| R-free | 0.223 (0.364) |
| Number of non-hydrogen atoms | 13,423 |
| macromolecules | 12,409 |
| ligands | 83 |
| water | 931 |
| Protein residues | 1,570 |
| RMS bonds (Å) | 0.007 |
| RMS angles (°) | 0.93 |
| Ramachandran favored (%) | 97.6 |
| Ramachandran allowed (%) | 22 |
| Ramachandran outliers (%) | 0.3 |
| Clash score | 6.0 |
| Average B-factor (Å$^2$) | 58.1 |
| macromolecules | 58.4 |
| ligands | 72.1 |
| solvent | 52.6 |

[a]Numbers in parentheses are statistics corresponding to the highest resolution shell.

prepared protein structures were solvated with TIP3P water molecules in a cubic periodic boundary system with a 10 Å cut-off to the next protein atom in each cartesian axis (final box dimensions 110.6 × 79.2 × 102.4 Å; 86,463 atoms). Ions were added to neutralize the systems. The resulting systems were relaxed using the MD-based Desmond Minimization protocol composed of 8 stages of a total duration of 160 ps. The production phase simulations were conducted using the OPLS_2005 force field in an NPT ensemble combined with the Martyna-Tobias-Klein barostat, with a relaxation time of 2.0 ps at 300 K, and the Nose-Hoover thermostat, with a relaxation time of 1.0 ps. The u-series algorithm was used to treat long-range interactions with a short-range interaction cutoff of 9 Å. By default, the M-SHAKE algorithm in Desmond was applied to constrain bonds to hydrogen atoms. We left the time step for the RESPA integrator at 2.0 fs, and files with atomic coordinates were saved at an interval of 48 ps. After the default relaxation protocol, the simulations were carried out in triplicate for a duration of 48 ns per ligand-protein complex at a temperature of 300 K, and to ensure a unique course of the individual trajectories, we generated random seeds for the initial velocities. The stability and convergence of the MD simulations were determined in the Simulation Interaction Diagram routine within Maestro.

The first 24 ns of each simulation were considered an equilibration phase and thus were not included in the subsequent analyses. The binding affinities were calculated using the MMGB/SA analysis based on 50 frames sampled between 24 and 48 ns of the MD simulation. An in-house script was used to evaluate per residue intramolecular as well as intermolecular hydrogen bonding and hydrophobic interactions of compstatin and its analogue with the C3b.

## Peptide synthesis

All peptides were synthesized using standard Fmoc chemistry on solid support (Rink amide resin) using a Liberty Blue peptide synthesizer (CEM, Kamp-Linfort, Germany) at 0.1 mmol scale in DMF. The following reagents were used: Fmoc-side chain protected amino acids (0.2 mM, 2.5 mL, 5 eq), Oxyma (1 mM, 0.5 mL, 5 eq), DIC (0.5 mM, 1 mL, 5 eq), 10% piperazine (in EtOH/NMP 1:9, 3 mL). The microwave-assisted cycles were designed as follows: after Fmoc deprotection (1 min, 90 °C) and three washing steps of 3 mL DMF each, amino acid coupling with Oxyma and DIC (4 min, 95 °C) was performed followed by one washing step (3 mL DMF). The obtained peptides were cleaved from the solid support with a cleavage cocktail of 92.5% TFA, 2.5% $H_2O$, 2.5% TIS, and 2.5% ethanedithiol for 3 h. The cleaved peptides were precipitated in 50 mL ice-cold ether, centrifuged (3215 × $g$, 5 min, 4 °C), and washed twice with 30 mL ice-cold ether. The crude peptides were dissolved in water (20 mL) to form the disulfide bonds by oxidation with $H_2O_2$; for this purpose, the pH of the solution was adjusted to pH 8 with 5% $NH_4OH$, and 3 eq 30% $H_2O_2$ was added. The reaction was stirred at room temperature for 30 min. The reaction progress was monitored by electrospray ionization mass spectrometry (ESI-MS; Micromass ZQ MAA 230; Waters, Milford, USA). After complete conversion to the disulfide peptides, the reaction was quenched with TFA and the peptides were lyophilized before purification by liquid chromatography mass spectrometry (LC-MS).

For purification, the crude cyclic peptides were dissolved in a mixture of $H_2O$/0.1% TFA and up to 20% ACN/ 0.1% TFA and were injected into an Infinity II 1260 semipreparative LC-MS instrument (Agilent, California, USA) using a C18 reverse phase column (XSelect, 250 × 19 mm, 5 μm; Waters), with the mobile phase as $H_2O$/ 0.1% TFA and ACN/0.1% TFA in a gradient (e.g., 20–40% ACN in 19.5 min). The peaks were detected by UV (214 nm and 254 nm) and mass. The purity of all peptides was accessed by analytical high-performance liquid chromatography (HPLC; Agilent 1100, Agilent, Santa Clara, USA) using a C18 column (Atlantis T3, 1 × 150 mm, 3 μm; Waters) and UV detection at 214 nm. The identity of all peptides was confirmed during LC-MS processing and the purity was calculated from the analytical HPLC (Supplementary Table 6, Supplementary Figs. 12 and 13).

Stock solutions for SAR studies were prepared by weighing the lyophilized purified fractions and determining the concentration by UV spectrophotometry (DS-11 spectrophotometer, DeNovix) at 280 nm using the corresponding extinction coefficients for each sequence.

A derivative of compstatin analog Cp05 containing a PEG$_2$ spacer (Fmoc-AEEAc-OH; AEE) at the N-terminus was synthesized as described above and coupled to both termini of a 40-kDa PEG moiety[36,45].

## Synthesis of N-alkylated Fmoc-L-tryptophan derivatives

The synthesis of N-alkylated Fmoc-L-tryptophan derivatives was achieved in five steps as described in the Supplementary Methods.

## Kinetic binding studies

All compstatin analogues were characterized concerning affinity and kinetic profiles by surface plasmon resonance (SPR) using a Biacore T200 instrument (Cytiva, Piscataway, USA). Experiments were run at 25 °C in HBST buffer (10 mM HEPES, 150 mM NaCl, 0.05% Tween-20, pH 7.4). C3b was immobilized on a carboxymethyl dextran hydrogel chip (CMD200; XanTec, Düsseldorf, Germany) using standard amine coupling in sodium acetate buffer pH 4.5 at a flow rate of 10 μl/min. Immobilization densities of 8000 and 4000 resonance units (RU) were achieved on separate flow cells. A reference flow cell was activated with the same method (NHS/EDC). All active sites were inactivated and blocked with ethanolamine. A single cycle approach at a flow rate of 30 μl/min was used for kinetic analysis by injecting five increasing

concentrations of each compound (0.5–40 nM) consecutively without regeneration in-between injections. Individual injections within a cycle were 120 s long with a 60 sec dissociation phase between injections. After the highest concentration of each derivative, a dissociation period of 6,000–14,400 sec was added to ensure complete dissociation in absence of suitable regeneration method. Analog Cp40 was included in each experimental series as an internal control. Signals from the reference cell from an ensemble of buffer blank injections were subtracted to correct for buffer effects and injection artefacts. Kinetic data was processed with the Biacore T200 Software (version 3.1; Cytiva) using a single-cycle evaluation setup by global fitting each data set to a 1:1 Langmuir binding model. Association and dissociation rate constants ($k_a$ and $k_d$, respectively) were fitted, and the equilibration constant $K_D$ was calculated based on the following equation: $K_D = k_d/k_a$. Each assay was performed at least twice and in duplicate injections.

### Solution binding studies

The interaction between soluble C3c and various compstatin analogs was assessed by isothermal titration calorimetry (ITC) using an $iTC_{200}$ instrument (MicroCal, Northampton, USA) at 25 °C using standard instrument settings (reference power 6 µcal s$^{-1}$, stirred speed 750 rpm, feedback mode high, filter period 2 s) and the $iTC_{200}$ control software. For each measurement, C3c was loaded to the sample cell at a concentration of 5 µM in PBS. Compstatin analogs in PBS were loaded to the syringe at concentrations of 50 µM (Cp01, Cp05, Cp40) or 25 µM (Cp05-PEG-Cp05; resulting in a 50 µM concentration of active entities) and injected to the sample cell at steps of 2 µl. Experiments were performed in duplicate and analyzed using AFFINImeter (v2.1802.5, Software for Science Developments, Santiago de Compostela, Spain).

### Functional assessment of convertase inhibition

The mode of inhibition of compstatin (Cp40) on C3 convertase was investigated by SPR using C3b deposited in a close-to-physiological manner[46]. For this purpose, initial C3b was immobilized by amine coupling (10 µg/ml in sodium acetate buffer pH 4.5) at low density (460 RU) on a carboxymethyl dextran chip (CMD500) at a flow rate of 10 µL/min. The buffer was subsequently substituted with HBST containing 1 mM $NiSO_4$, and more C3b was deposited subsequentially by generating the C3 convertase on the chip by injecting premixed FB and FD (100 nM each) followed by 500 nM C3 multiple times till 3090 RU of C3b were reached. The reference flow cell was activated and deactivated as described above.

Experiments were run at 25 °C in HBST buffer containing 1 mM $MgCl_2$ at 10 µL/min. Signals were referenced as described above and processed in the Biacore software (Cytiva) and Scrubber (version 2.0c, BioLogic, Campbell, Australia). To assess convertase formation and activity, an initial injection of buffer (no convertase) or premixed FB/FD mixture (100 nM each; convertase) for 180 s with 100 s dissociation was directly followed by an injection of C3 (250 nM, 120 s contact time, 120 s dissociation). In the first cycle only, C3 without previous convertase formation was injected, while for cycles 2 and 3, FB/FD and C3 were injected. While cycles 1 and 2 were run in normal HBST + $MgCl_2$ buffer, the third cycle was run in HBST + $MgCl_2$ buffer containing 250 nM Cp40.

To evaluate whether the binding of Cp40 to the substrate (C3) or the convertase (C3bBb) primarily contributes to the protein-protein interaction inhibition, SPR experiments using C3b deposited in a close-to-physiological manner as described above were conducted with a mutant of FB, which contained a Ser-to-Ala mutation in the catalytic site to render it inactive[5]. Experiments were run at 25 °C in HBST buffer containing 1 mM $NiCl_2$ at 10 µL/min. Signals referenced and processed as described above. An initial injection of premixed FB*/FD mixture (100 nM each; convertase) for 240 s with 180 s dissociation was directly followed by an injection of C3 (500 nM, 60 s contact time, 180 s dissociation), Cp40 (500 nM, 60 s contact time, 0 s dissociation)

C3 (500 nM, 60 s contact time, 180 s dissociation), and a co-incubated 1:1 mixture of Cp40 and C3 (500 nM each, 60 s contact time, 180 s dissociation). To show the concentration dependence increasing amounts of Cp40 preincubated with C3 (500 nM) were injected.

The interaction of C5 with the C3bBb complex was performed as described for the experiments with C3 above. The C3 convertase was formed by injecting FB and FD on a C3b-coated sensor chip and C5 (500 nM) was injected to measure the interaction. The same injection cycle was repeated in regular running buffer and in running buffer containing Cp40.

### Determination of species specificity

The inhibitory efficacy of Cp40 on complement activation (classical pathway) in plasma of different species was measured by ELISA[11,47]. Plasma samples from different species (human, pig, dog, rabbit, rat, and mouse), collected in EDTA, were reconstituted for complement activity by diluting them 1:80 in veronal buffer (5 mM barbital, 145 mM NaCl, 0.1% gelatin, pH 7.4) containing $CaCl_2$ and $MgCl_2$. Plates were coated with 1% chicken ovalbumin for 2 h, blocked with 1% BSA (1 h), and incubated with a polyclonal rabbit anti-chicken ovalbumin antibody (1:1000; Cocalico) for 1 h at room temperature. After washing, reconstituted plasma samples containing increasing concentrations of Cp40 (0.04–5 µM) were added to the plates, incubated for 1 h at room temperature, and washed with veronal buffer containing 0.5 mM $MgCl_2$ and 0.15 mM $CaCl_2$. Complement activation was determined by detecting the deposition of C3 activation fragments using HRP-conjugated anti-C3 antibodies with appropriate species reactivity. The data were corrected for background activation and divided by the signal in absence of Cp40 to achieve percent inhibition values.

### Homology model of mouse C3b

A homology model of the β-chain of mouse C3b was built using the SwissModel platform (https://swissmodel.expasy.org)[48]. The primary sequence of residues 25–666 of mouse C3 (Uniprot P01027) in FASTA format was used as input for the SwissModel web server. Ten homology models were built based on the best-matching templates identified by sequence similarity to existing proteins. All homology models were superimposed on the human C3b-Cp40 complex and visually inspected. The top scored model (100% residue coverage, 76% sequence identity, GMQE score 0.87, QMEAN score 0.06) built on the template of human C3b-MCP (PDB 5FO8, 2.4 Å) was used for subsequent modeling. After alignment, side chain rotamers within the binding site of the best homology model were manually adjusted to mimic rotamers of the human C3b-Cp40 structure. Cp40 was placed in the homology model by rigid body transfer, and the complex was optimized using restrained minimization (converging heavy atoms to RMSD: 1.0 Å) using the OPLS_2005 force field. Structure alignment and refinement was performed in Maestro (v10.4, Schrödinger, New York). The Protein Preparation Wizard minimizer routine with OPLS_2005 force field was used to relax manually constructed protein-ligand complexes, and Maestro for visualization[49].

### Reporting summary

Further information on research design is available in the Nature Research Reporting Summary linked to this article.

## Data availability

The data that support this study are available from the corresponding author upon request. Atomic coordinates for the reported crystal structure have been deposited in the RCSB Protein Data Bank under accession number 7BAG (C3b-Cp40). The following structures used to perform analyses and/or generate figures are available under the following accession numbers: 2A73, 2I07, 2OK5, 2WII, 2WIN, 2XW9, 4HW5, 5FO8. Source data are provided with this paper.

## Code availability

A computer program (source code) used to evaluate the per-residue intramolecular and intermolecular hydrogen bonding and hydrophobic interactions of C3b with Cp40 and analogues thereof has been deposited on GitHub (https://github.com/mmodbasel/CP40_SAR, https://doi.org/10.5281/zenodo.6839171).

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

## Acknowledgements

We gratefully thank the European Synchrotron Radiation Facility (ESRF) for the provision of synchrotron radiation facilities and beamline scientists of the ESRF and the European Molecular Biology Laboratory for assistance. This study was supported by grants from the Swiss National Science Foundation (31003A_176104 to D.R.; 205321_204607 to M.S. and D.R.), ZonMW (Top grant 700.54.304 to P.G.), the Netherlands Organization for Scientific Research (01.80.104.00 to P.G.), the European Community's Seventh Framework Programmes (FP7-IDEAS 233229 to P.G.; FP7-INFRASTRUCTURES 283570 to P.G.) and the U.S. National Institutes of Health (P01AI068730 to J.D.L.), and by the Ralph and Sallie Weaver Professorship in Research Medicine (to J.D.L.). The authors thank Butler SciComm for their excellent language editing service.

## Author contributions

C.L. and N.B. synthesized monovalent and bivalent compstatin analogs, respectively. C.L., J.D.L. and D.R. designed, performed, and interpreted the interaction studies. X.G., H.v.S and P.G. crystallized C3b-Cp40 and prepared, evaluated, and interpreted the structural data. M.S. performed molecular dynamics simulations, analyzed interactions in silico, and developed the homology model of mouse C3b. B.W. synthesized N-alkylated Fmoc-Trp-OH derivatives. G.S. performed species specificity assays. C.L., X.G., M.S., and D.R. wrote the manuscript. D.R., J.D.L. and P.G. supervised the studies. All authors contributed to editing and critical proof-reading of the paper.

## Competing interests

J.D.L. is the founder of Amyndas Pharmaceuticals, which is developing complement inhibitors for therapeutic purposes, is the inventor of patents or patent applications that describe the use of complement inhibitors for therapeutic purposes (patent WO/2013/036778 relevant to this study), some of which are being developed by Amyndas Pharmaceuticals, is the inventor of the compstatin technology licensed to Apellis Pharmaceuticals (Cp05/POT-4/APL-1 and PEGylated derivatives such as APL-2/pegcetacoplan and APL-9), and has provided paid consulting services to Achillion, Ra Pharma, Viropharma, Sanofi, Shire, LipimetiX and Baxter. D.R. is the inventor of patents or patent applications that describe complement inhibitors for therapeutic purposes (patent WO/2013/036778 relevant to this study) some of which are developed by Amyndas Pharmaceuticals, has provided paid consulting services to Roche Pharma, Sobi and Greenovation, and provided scientific lectures sponsored by Roche and Alexion. The other authors claim no conflict of interest.
