## [Peer Review File · Nature Communications]

Insight into mode-of-action and structural determinants of the compstatin family of clinical complement inhibitorsREVIEWER COMMENTS

Reviewer #1 (Remarks to the Author):

The manuscript of Lamers et al describes the crystal structure of the complement C3 inhibitor CP40 with its ligand C3b, explaining its strong affinity and species specificity. The authors provide also some novel insights on the action of CP40 on the alternative pathway C3 convertase. The presented results are novel and original and of a great interest for the community, since CP40 is in clinical trials and detailed knowledge of its mode of action is necessary in order to find the best indications. The methodology is sound and well described. The text is well written and easy to follow. What needs improvement is the dissection of the mechanism of action of CP40 on the convertases. Only limited SPR data are shown to explain that CP40 prevent the initial binding of the substrate C3 to the alternative pathway C3 convertase C3bBb.

What is important to show is whether and how CP40 acts on the classical pathway C3 convertase, which also uses C3 as a substrate. I.e., if C3 is loaded with CP40, will it be cleaved by the classical pathway convertase? Also, how it acts on the C5 convertases? They contain C3b, therefore one can expect that loading of this C3b with CP40 will affect the terminal pathway as well, even if the C5 convertase is already existing. These questions can be answered by SPR assays, forming the C3 convertase on the chip (classical or alternative) and flowing CP40-loaded or not C3. Alternatively, the convertase on the chip (C3 or C5) can be loaded with CP40 after formation and C5 can be injected to follow if it will bind. Further, experiments with beads loaded with C3b can be used in solution in similar settings in order to measure C3a and C5a in the fluid phase as a readout of the activity of each of the convertases. Alternatively, hemolytic tests are also an excellent option.

On page 10, line 298-299 there is a sentence which states that complement activation process requires continues convertase formation to attract a relevant amount of C3 to the target surface. It is unclear from which result this conclusion is made. It will be better to remove it from the results section and use it in the discussion, providing reference to the exact result to which it refers to (is it presented in the manuscript?).

Reviewer #2 (Remarks to the Author):

This research paper is sounded and rooted on a top level and extensive multidisciplinary approach. The quality of the figures is very much appreciated, although legends or labels should be carefully checked.

This study mainly experimentally explores and analyses the molecular determinants of the functional improvement of the compstatin family of complement inhibitors (Figure 1), namely the current clinical candidate Cp40 version as compared to the original compstatin (Cp01) inhibitor version from which it derives.

The comparative analysis of Cp01 and Cp40 interactions with C3b/c are first dissected through comparison of the crystal structures of their complex with C3c and C3b, respectively (Figure 2). This is followed by an exhaustive molecular dynamics approach which enables to estimate the binding energies of each residue and escape the possible bias of local conformations selected by crystal packing (cf Figure S5). This molecular dynamic study analyses both the C3b binding contributions (Figure 3a, 3b) and intramolecular stabilization (Figure 3c, 3d). Close up views are provided and commented for further important details (Figure 3e-j). To further validate the previous analysis results, comprehensive structure-activity relationship studies were performed, in order to analyze the isolated effect of each substitution between Cp01 and Cp40 (Figure 4), which have contributed to such an overall 300-fold affinity increase. This helps to more precisely dissect the contribution of each inhibitor position in the interaction. The study highlights that the major impacts on interaction with C3b are obtained for the combined mutation of charged residues R12 and D7 (Figure 4d), and for the mutation at position 5 (Figure 4c and S7, Table 1).

Then, coming to the question of the functional mechanism, the investigators ask whether Cp40 mainly impacts the C3 convertase formation or other interactions, as nicely illustrated on Figure 5a. The main interpretation proposed here is that Cp40 binding to C3 will prevent C3 from binding to the C3 convertase and thus get cleaved into C3a and C3b, a crucial step of the amplification loop (Figures 5b, S8). Indeed, it is elegantly shown that Cp40 does not prevent the initial C3 convertase formation (Figure 5c), but its effect is seen when C3 is then added to the active or inactive C3bBb convertase (Figure 5c, 5e).

The last part addresses the question of Cp40 species specificity of Cp40, since it only inhibits complement activation in human plasma (Figure 6a). This property sounds consistent with the previous analysis of Cp40 binding site, which is more conserved in non-human primate C3 than in other species (Figure S10c).

Altogether, these data provide rational clues to explain at the molecular level how recent compstatin-derived inhibitors such as Cp40 gained improved (300-fold) affinity towards human C3-like molecules, and at which level they inhibit the complement pathway, i.e. at the level of active C3 recruitment to the C3 convertase, which is a significant advance. This paves the way for further improvement or development of analogs. The question of a possible C3 bypass remains open (lines 434-439) for certain diseases (but not PNH).

MAJOR COMMENTS:

The last part about species specificity and using the mice C3 models is the weaker part of the manuscript. This part could probably be improved, clarified or simplified. It is indeed a complex subject lacking the same level of experimental support as the previous parts. These data do not compromise the interest of the manuscript but might be shifted to the discussion as a support and consistent outcome of the non-conservation (e.g. in rodents) of the previously defined Cp40 binding site in human C3 and derivatives.

If you really need to keep this part, at least improve the legend for Fig. 6 b,c (and correct for 6d instead of 6b). The reader will need more help to understand what Figure 6 b-c aims to show. From the text, the reader would guess that Figure 6b is the mouse model but then the labels are misleading. The labels shown indeed correspond to human C3 (e.g. His392-Pro393), so what are the structures shown (-> no mouse model)? The actual Fig. 6c seems quite redundant with Fig. 3f, but so many dashed lines may rather suggest clashes???? No structural water molecule there, come back to this question?

OTHER REQUIRED IMPROVEMENTS:

Lines 346-7 limited in this sentence sounds to me somehow potentially misleading . Do you mean strictly limited to human C3 (and probably non-human primates)?

Line 373: C3b is ONE of the natural targets of Cp40

CAREFULLY CHECK THE LEGENDS OF FIGURE AND TABLES:

Legend of Figure 3a could be improved. H-bond yellow, not red; dark shade not easy to understand at first sight.

Please use a black label for the black curve in Fig4a,c,d,e.

Fig. 4e is great but lacks references in the text.

Correct the red and blue labels which I guess have been exchanged in Fig 5c.

Table 1 legend: would make sense to cite the nature of the ligand tested (C3b, I assume)

Minor COMMENTS:

Short title could be improved

Fig S2a 2Fo-Fc is slightly biased towards the model... an omit map would be more convincing.

Fig S8 cyan label = C3 (below 2A73)

Figure S1 highlights that the CUB domain is less defined in the present crystal structure as compared to previous ones with different crystal packing. How does the CUB module behave during MD studies?

Line 103: THE latter (?)

So many auto-citations of previous supervisors papers! Could be slightly more equilibrated on this side...

Point-by-point response to reviewer comments

Reviewer #1 (Remarks to the Author):

The manuscript of Lamers et al describes the crystal structure of the complement C3 inhibitor CP40 with its ligand C3b, explaining its strong affinity and species specificity. The authors provide also some novel insights on the action of CP40 on the alternative pathway C3 convertase. The presented results are novel and original and of a great interest for the community, since CP40 is in clinical trials and detailed knowledge of its mode of action is necessary in order to find the best indications. The methodology is sound and well described. The text is well written and easy to follow.

A: We thank the reviewer for the positive assessment of our study and the constructive comments that helped to improve the manuscript during the revision.

What needs improvement is the dissection of the mechanism of action of CP40 on the convertases. Only limited SPR data are shown to explain that CP40 prevent the initial binding of the substrate C3 to the alternative pathway C3 convertase C3bBb.

What is important to show is whether and how CP40 acts on the classical pathway C3 convertase, which also uses C3 as a substrate. I.e., if C3 is loaded with CP40, will it be cleaved by the classical pathway convertase? Also, how it acts on the C5 convertases? They contain C3b, therefore one can expect that loading of this C3b with CP40 will affect the terminal pathway as well, even if the C5 convertase is already existing. These questions can be answered by SPR assays, forming the C3 convertase on the chip (classical or alternative) and flowing CP40-loaded or not C3. Alternatively, the convertase on the chip (C3 or C5) can be loaded with CP40 after formation and C5 can be injected to follow if it will bind. Further, experiments with beads loaded with C3b can be used in solution in similar settings in order to measure C3a and C5a in the fluid phase as a readout of the activity of each of the convertases. Alternatively, hemolytic tests are also an excellent option.

A: Considering the already substantial scope of the study and the size limitations by the publisher, we had opted to focus the mechanistic studies to the C3-C3bBb system but agree that extending data and/or discussions on the mode of action would be helpful. We therefore extended our SPR studies to further dissect the mechanism of Cp40 on the alternative pathway C3 convertase. In particular, we could show that both the binding of Cp40 to the C3 substrate and to C3b as part of the C3bBb complex contribute to the inhibitory activity.

We also added direct binding data on the impact of Cp40 on the C5-C3bBb interaction. In analogy to previous studies mentioned in the original manuscript (Mannes et al., 2021; PMID 33507296), we observed only a low interference of Cp40 in those binding events (Fig. 5h). While important, it is difficult to extrapolate from such studies to the potential impact on the CP/LP C3 convertase or any C5 convertase. Our studies in this manuscript have shown that the binding of C3 to immobilized free C3b and C3bBb is highly distinct and the impact therefore much more relevant for the C3-C3bBb interaction. In principle, the same experiment had to be repeated with an on-chip C4b2b convertase complex. Unfortunately, the assembly of such convertases is highly challenging when compared to the AP C3 convertase and, to our knowledge, there are no published protocols (Mannes et al. also only use the AP C3 convertase for SPR). Despite several efforts during the preparation of this revision, our group also did not manage to reproducibly generate a CP/LP C3 convertase that would allow for reliable mechanistic studies.

The situation is even more limited in the case of C5 convertases, the exact composition of which is still matter of investigation and debate. Among the favored hypotheses is that high C3b densities would impose a conformational change in C5 that facilitates its cleavage by a convertase (likely even by C3bBb). The newly added data in Fig. 5 show that C5 indeed binds to C3bBb and that Cp40 leads to a partial reduction of

this interaction. When considering the strong effect of Cp40 on C3b deposition, we still assume that the impact of the inhibitor on C5 effectors is mediated by reduced opsonization rather than a direct effect on C5 activation.

Addressing the exact mode-of-action of compstatin analogs for CP/LP C3 convertases and “C5 convertases” on a molecular level would therefore entail the development and validation of completely novel assay formats, which would exceed the scope of the current study (though we certainly aim to contribute to answering such questions in the future).

Given that a direct comparison of the effects of Cp40 on the various convertases is currently not possible in comparable molecular detail and considering that the inhibitory effect on Cp40 and other compstatin derivatives on CP-mediated C3b activation/deposition and hemolysis has been shown in numerous published studies (e.g., PMID 34526511, 31171642, 29960275, 29920096, 27677785, 26548839, 25350518, 22851705, 22795972, 21067811, 18957045, 18431241, 18424758, 17101176, 16854067, 12902490, 10355733, and 8752942, among others), we did not extend the experimental panel on this aspect but rather emphasized the limitations, published data and open questions.

On page 10, line 298-299 there is a sentence which states that complement activation process requires continuous convertase formation to attract a relevant amount of C3 to the target surface. It is unclear from which result this conclusion is made. It will be better to remove it from the results section and use it in the discussion, providing reference to the exact result to which it refers to (is it presented in the manuscript?).

A: We fully agree with the reviewer that this statement fits much better to the discussion section as it is not directly derived from the results but rather based on its context with published literature.

Reviewer #2 (Remarks to the Author):

This research paper is sounded and rooted on a top level and extensive multidisciplinary approach. The quality of the figures is very much appreciated, although legends or labels should be carefully checked. This study mainly experimentally explores and analyses the molecular determinants of the functional improvement of the compstatin family of complement inhibitors (Figure 1), namely the current clinical candidate Cp40 version as compared to the original compstatin (Cp01) inhibitor version from which it derives. The comparative analysis of Cp01 and Cp40 interactions with C3b/c are first dissected through comparison of the crystal structures of their complex with C3c and C3b, respectively (Figure 2). This is followed by an exhaustive molecular dynamics approach which enables to estimate the binding energies of each residue and escape the possible bias of local conformations selected by crystal packing (cf Figure S5). This molecular dynamic study analyses both the C3b binding contributions (Figure 3a, 3b) and intramolecular stabilization (Figure 3c, 3d). Close up views are provided and commented for further important details (Figure 3e-j). To further validate the previous analysis results, comprehensive structure-activity relationship studies were performed, in order to analyze the isolated effect of each substitution between Cp01 and Cp40 (Figure 4), which have contributed to such an overall 300-fold affinity increase. This helps to more precisely dissect the contribution of each inhibitor position in the interaction. The study highlights that the major impacts on interaction with C3b are obtained for the combined mutation of charged residues R12 and D7 (Figure 4d), and for the mutation at position 5 (Figure 4c and S7, Table 1). Then, coming to the question of the functional mechanism, the investigators ask whether Cp40 mainly impacts the C3 convertase formation or other interactions, as nicely illustrated on Figure 5a. The main interpretation proposed here is that Cp40 binding to C3 will prevent C3 from binding to the C3 convertase and thus get cleaved into C3a and C3b, a crucial step of the amplification loop (Figures 5b, S8). Indeed, it is elegantly shown that Cp40 does not prevent the initial C3 convertase formation (Figure 5c), but its effect is seen when C3 is then added to the active or inactive C3bBb convertase (Figure 5c, 5e). The last part addresses the question of Cp40 species specificity of Cp40, since it only inhibits complement activation in human plasma (Figure 6a). This property sounds consistent with the previous analysis of Cp40 binding site, which is more conserved in non-human primate C3 than in other species (Figure S10c). Altogether, these data provide rational clues to explain at the molecular level how recent compstatin-derived inhibitors such as Cp40 gained improved (300-fold) affinity towards human C3-like molecules, and at which

level they inhibit the complement pathway, i.e. at the level of active C3 recruitment to the C3 convertase, which is a significant advance. This paves the way for further improvement or development of analogs. The question of a possible C3 bypass remains open (lines 434-439) for certain diseases (but not PNH).

A: We very much appreciate the reviewer's detailed and positive feedback on the findings of our study and for the helpful comments and suggestions.

MAJOR COMMENTS:

The last part about species specificity and using the mice C3 models is the weaker part of the manuscript. This part could probably be improved, clarified or simplified. It is indeed a complex subject lacking the same level of experimental support as the previous parts. These data do not compromise the interest of the manuscript but might be shifted to the discussion as a support and consistent outcome of the non-conservation (e.g. in rodents) of the previously defined Cp40 binding site in human C3 and derivatives. If you really need to keep this part, at least improve the legend for Fig. 6 b,c (and correct for 6d instead of 6b). The reader will need more help to understand what Figure 6 b-c aims to show. From the text, the reader would guess that Figure 6b is the mouse model but then the labels are misleading. The labels shown indeed correspond to human C3 (e.g. His392-Pro393), so what are the structures shown (-> no mouse model)? The actual Fig. 6c seems quite redundant with Fig. 3f, but so many dashed lines may rather suggest clashes???? No structural water molecule there, come back to this question?

A: We agree with the reviewer and have revised Figure 6 and the corresponding labels to further improved the clarity.

OTHER REQUIRED IMPROVEMENTS:

Lines 346-7 limited in this sentence sounds to me somehow potentially misleading . Do you mean strictly limited to human C3 (and probably non-human primates)?

A: We have revised the statement accordingly.

Line 373: C3b is ONE of the natural targets of Cp40

A: We thank the reviewer for raising our awareness about this oversight. The phrase has been revised.

CAREFULLY CHECK THE LEGENDS OF FIGURE AND TABLES:

Legend of Figure 3a could be improved. H-bond yellow, not red; dark shade not easy to understand at first sight.

Please use a black label for the black curve in Fig4a,c,d,e. Fig. 4e is great but lacks references in the text.

Correct the red and blue labels which I guess have been exchanged in Fig 5c.

Table 1 legend: would make sense to cite the nature of the ligand tested (C3b, I assume)

A: Thank you very much for pointing this out. All labels and legends have been checked and corrected.

Minor COMMENTS:

Short title could be improved

A: Any short titles added to previously submitted documents have only served to guide the reviewers. Nature Communication does not include short titles for their published articles.

Fig S2a 2Fo-Fc is slightly biased towards the model... an omit map would be more convincing.

A: An omit map has been added to Fig. S2 as new panel b. The electron density map (a) and b-factors (c) have been updated to match the view of panel b.

Fig S8 cyan label = C3 (below 2A73)

A: This has been changed.

Figure S1 highlights that the CUB domain is less defined in the present crystal structure as compared to previous ones with different crystal packing. How does the CUB module behave during MD studies?

A: In Fig. S1s, we had demonstrated that the CUB domain in C3b-CP40 structure does not contribute any crystal contacts, providing a rationale for the poor electron density of the CUB. The flexibility is a natural feature of CUB domains as the conformation of R1303 and S1304 in C3b CUB needs to translocate in order to allow the cleavage by Factor I [PMID 28671664]. Even in cases where they contribute to crystal contacts, the b-factors of CUB domains in C3b structures are high [27013439].

Furthermore, the CUB domain is located in a substantial distance (~60 Å) to the binding site of Cp40 and thereby not involved in the drug-target interaction. We had therefore omitted the CUB from the model used in our MD simulation and therefore cannot assess the stability of behavior of this domain during MD simulations.

Line 103: THE latter (?)

A: The typo has been corrected.

So many auto-citations of previous supervisors papers! Could be slightly more equilibrated on this side...

A: Owing to the central role of Dr. John Lambris (who is not only a former supervisor but co-corresponding author in this study...;-) as inventor and developer of the compstatin technology and his long-standing contributions to the field, a certain numeric surplus of cited publications from this author is determined by the topic. That said, we replaced some of the citations on non-compstatin-related statements to equilibrate the bibliography.

REVIEWER COMMENTS

Reviewer #1 (Remarks to the Author):

The authors addressed all my comments

Reviewer #2 (Remarks to the Author):

The revised version has been significantly improved and is now mature for publication. Congratulations to the authors for this work.